# EvoWorld: Evolving Panoramic World Generation with Explicit 3D Memory

## Abstract

Humans possess a remarkable ability to mentally explore and replay 3D environments they have previously experienced. Inspired by this mental process, we present EvoWorld: a world model that bridges panoramic video generation with evolving 3D memory to enable spatially consistent long-horizon exploration. Given a single panoramic image as input, EvoWorld first generates future video frames by leveraging a video generator with fine-grained view control, then evolves the scene's 3D reconstruction using a feedforward plug-and-play transformer, and finally synthesizes futures by conditioning on geometric reprojections from this evolving explicit 3D memory. Unlike prior state-of-the-arts that synthesize videos only, our key insight lies in exploiting this evolving 3D reconstruction as explicit spatial guidance for the video generation process, projecting the reconstructed geometry onto target viewpoints to provide rich spatial cues that significantly enhance both visual realism and geometric consistency. To evaluate long-range exploration capabilities, we introduce the first comprehensive benchmark spanning synthetic outdoor environments, Habitat indoor scenes, and challenging real-world scenarios, with particular emphasis on loop-closure detection and spatial coherence over extended trajectories. Extensive experiments demonstrate that our evolving 3D memory substantially improves visual fidelity and maintains spatial scene coherence compared to existing approaches, representing a significant advance toward practical long-horizon spatially consistent world modeling.

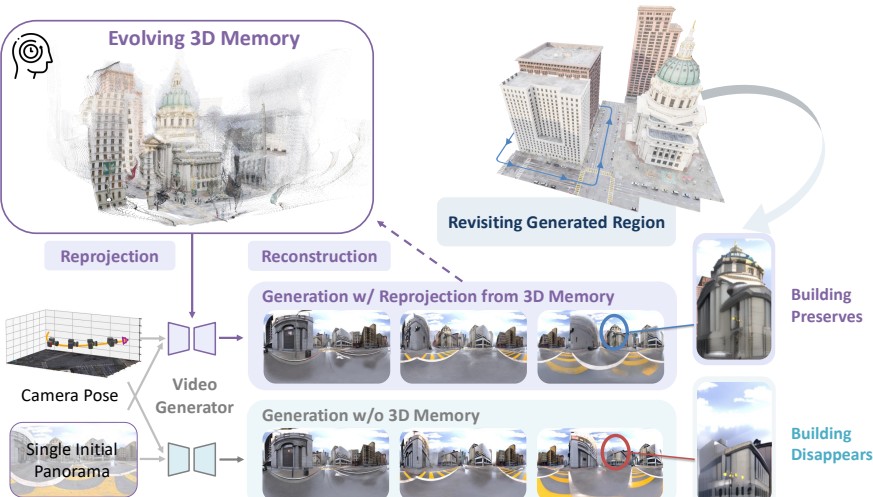

Figure 1: Panoramic world generation with explicit reconstructed 3D memory. Our EvoWorld continually reconstructs a 3D memory based on previously generated frames and generates spatial-consistent video. The vanilla generator without memory (Lu et al., 2025) exhibits spatial inconsistency.

# 1 INTRODUCTION

Humans effortlessly build mental world models (Johnson-Laird, 1983): after a single glance, they infer a coherent 3D scene, imagine occluded regions, and maintain a consistent internal map while moving. This remarkable ability has long been a challenge in the development of artificial intelligence.

Recent advances in large video generative models (OpenAI, 2024; Blattmann et al., 2023a; Agarwal et al., 2025) have shown the capability of generating high-quality video from image and text prompts. Interactive generative world models (Van Hoorick et al., 2024; Lu et al., 2025) extend the trend: given the agent's current partial view of the world and a candidate action, the model predicts the resulting video, serving as a realistic world model. This suggests a promising path towards building a computational analog of human mental world models, serving as the simulator of the physical world.

However, maintaining 3D consistency over time continues to be a fundamental hurdle. The problem becomes even more evident when generating long video sequences, especially in looping scenarios where the camera returns to a previously visited location. During long or looping trajectories, geometry drifts: an agent revisiting a location (Fig. 1) finds its surroundings altered. This reveals a critical gap: the generated scene can change arbitrarily without an explicit memory of the environment.

To narrow the gap, we propose EvoWorld, a generative world model with explicit 3D reconstructed memory. Unlike conventional approaches that generate frames independently or rely solely on short-term dependencies, EvoWorld explicitly reconstructs and updates a 3D point cloud representation of the scene, which we term *explicit 3D memory*. This *explicit 3D memory* provides structural priors that help maintain 3D consistency even over extended sequences.

Our key insight is to use an *explicit 3D memory* as a guiding prior for video generation. As new frames are synthesized, this memory is dynamically updated via VGGT (Wang et al., 2025) in a feed-forward manner and projected onto future viewpoints to condition the generator. To enable fine-grained view control in 360° panoramic generation, we further introduce a spherical Plücker embedding to encode camera parameters. By explicitly incorporating the spatial cues reprojected from 3D memory, EvoWorld enhances realism, reduces temporal inconsistencies, and enables more controllable scene exploration. Crucially, EvoWorld alleviates the drifting problem, ensuring that previously visited locations remain consistent even when revisited later in the sequence.

We curate the first thorough dataset, Spatial360, for long-range and looping exploration across simulated outdoor (Unity and UE5), Habitat indoor, and real-world scenes. We demonstrate that EvoWorld significantly outperforms state-of-the-art methods in both visual fidelity and 3D consistency.

Our contributions are summarized as follows:

- We propose a framework for evolving world generation that builds and updates an explicit 3D memory from generated videos. The reconstructed 3D geometry mitigates error accumulation, ensuring spatial consistency over time.
- We enable fine-grained view control for 360-degree panoramic generation using a spherical Plücker embedding that encodes camera parameters.
- We introduce Spatial360, a high-quality open dataset of panoramic videos and camera poses spanning synthetic outdoor, Habitat indoor, and real-world environments. This dataset facilitates research on long-range exploration and loop closure in both simulated and real-world settings.

By integrating explicit reconstructed memory with generative video synthesis, EvoWorld opens new possibilities for producing 3D-consistent world exploration. We also demonstrate that EvoWorld can be helpful for downstream tasks such as target reaching. We believe this framework and the dataset collectively establish a solid foundation for future research on scalable, physically grounded generative world models with explicit 3D memory.

# 2 METHOD

We introduce a memory-augmented world explorer that incrementally reconstructs a 3D map of its surroundings and generates new frames conditioned on this evolving 3D representation. By recalling and integrating relevant spatial information, our approach improves the spatial consistency of generated videos, leading to greater temporal coherence and realism. Section 2.1 introduces

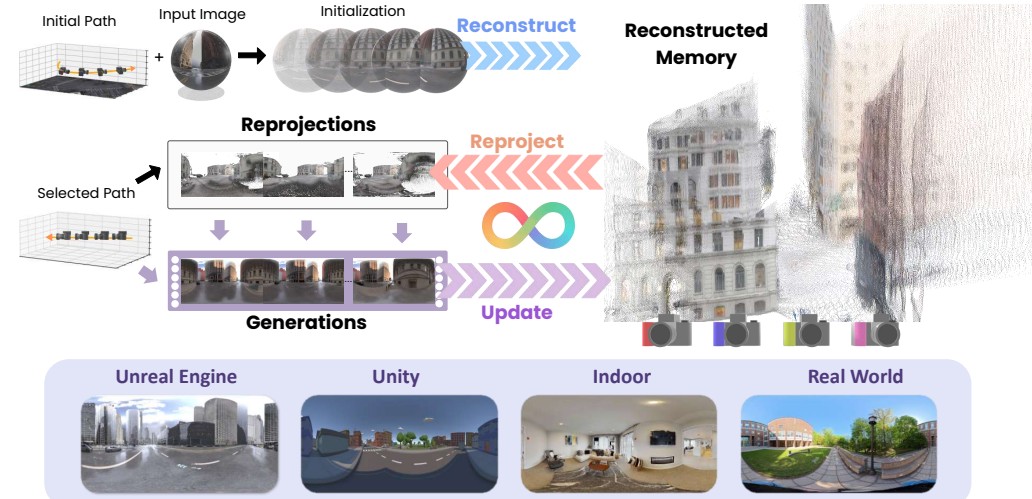

Figure 2: Overview of EvoWorld. Starting from a single panoramic frame and view control, EvoWorld generates spatially consistent videos by iteratively alternating between 3D reconstruction and video generation, where the video generation is conditioned on reprojections from the evolving 3D memory.

the fundamentals of diffusion models for panoramic video generation. Section 2.2 describes our 3D memory representation and the process of reconstruction and view-projection. Section 2.3 explains how the 3D memory and camera control signals are integrated into the generative model, and Section 2.4 shows how generation and reconstruction are performed iteratively. The overall framework is illustrated in Figure 2, and the inference process is summarized in Algorithm 1.

## 2.1 PRELIMINARY

**World models** provide predictive representations of future states by modeling the probabilistic distribution of state transitions $p(x_{t+1}|x_t, a_t)$, given the current observation $x_t$ and action $a_t$.

**Generative world explorer** (GenEx) (Lu et al., 2025) grounds the world models on the realistic physical world, differentiating from the early attempts of world models on simple game agents. An explorable generative world aims to dynamically expand by generating a video conditioned on an agent's immediate surroundings. Specifically, given an initial panoramic image $x_0$, we use a conditional video diffusion model to synthesize temporally coherent video sequences.

**Recursive video generation**. When using diffusion models to generate longer videos autoregressively, a common strategy is to first synthesize a short video clip. This clip then serves as a foundation for iterative extension, where each subsequent segment is generated by conditioning on the previously produced segment. This procedure enables the construction of longer videos while preserving temporal coherence and consistency. Suppose in each autoregressive step $t$, $S + 1$ video frames are generated. Starting from a single panoramic image $x_0$, we define the generated panoramic video clip at exploration step $t$ as $\mathbf{x}_t = (x_t^0, x_t^1, \ldots, x_t^S)$, where $x_t^S$ is the latest explored panoramic view, and $x_{t+1}^0 := x_t^S$, thus we can generate videos recursively by conditioning on the last frame in the previous exploration step.

## 2.2 EXPLICIT 3D MEMORY REPRESENTATION

**Memory representation.** Given the previously generated panoramic videos $\mathbf{x}_{0:t-1}$, we convert them into cubemaps (see Appendix), and reconstruct a 3D world $\mathcal{M}_t$ representing the explored environment up to step $t$. $\mathcal{M}_t$ can take various forms, e.g, point clouds, meshes, NeRFs, or Gaussian splats, that support novel view synthesis. We adopt colored point clouds for their simplicity and efficiency, using a feed-forward network to infer 3D attributes. Recent methods (Wang et al., 2025; Yang et al., 2025) enable real-time reconstruction, making the overhead negligible. Specifically, at step $t$, given an action $\mathbf{a}_t \sim \mathcal{A}$, we can get the camera locations (panorama center) $\boldsymbol{\delta}_t$ and rotations (forward-facing direction) $\boldsymbol{\alpha}_t$, which are the target views associated with the $\mathbf{x}_t$.

---

**Algorithm 1** Generating an Explorable World with Explicit 3D Memory $p(\mathbf{x}_{0:T} \mid x_0, \mathbf{a}_{0:T})$

---

**Require:** • An initial panoramic image $x_0$.
  • Action space $\mathcal{A}$ defined in the physical engine, from which an action is sampled: $\mathbf{a}_t \sim \mathcal{A}$. $\mathbf{a}$ can be a camera control.
  • A conditional distribution $p_\theta(\mathbf{x} \mid x_{t-1}^S, \mathbf{a}_t)$, parameterized by a panoramic video generation model $\theta$.
  • A memory base $\mathcal{M}_t$ that stores the reconstructed 3D point cloud using previous generated videos $\mathbf{x}_{0:t-1}$.
1: Notation: Let $\mathbf{x}_t = (x_t^0, x_t^1, \ldots, x_t^S)$ denote the generated panoramic video at exploration step $t$. Each step contains $S + 1$ frames. $x_t^S$ is the latest explored panoramic view, and $x_{t+1}^0 := x_t^S$, thus we can generate videos recursively by conditioning on the last frame in the previous exploration step.
2: **World initialization**: Initialize a $360°$ panoramic world from a single panoramic image $x_0$.
3: **for** $t = 1$ to $T$ **do**
4:   **World transition** at step $t$: Given an action $\mathbf{a}_t \sim \mathcal{A}$, update camera locations $\boldsymbol{\delta}_t$ and rotations $\boldsymbol{\alpha}_t$ associated with the $\mathbf{x}_t$. Reproject the 3D point cloud from the memory base $\mathcal{M}_t$ onto the image planes based on $\boldsymbol{\delta}_t$ and $\boldsymbol{\alpha}_t$, denoted as $\mathbf{r}_t$. Given the above and the latest explored world $x_{t-1}^S$ (where $x_0^S := x_0$), we can sample the new panoramic video $\mathbf{x}_t$:

$$\mathbf{x}_t \sim p_\theta(\mathbf{x} \mid x_{t-1}^S, \boldsymbol{\delta}_t, \boldsymbol{\alpha}_t, \mathbf{r}_t)$$

5:   **Memory Base Update**: Incorporate the newly generated video into the reconstruction function $\mathcal{R}$ and update the 3D point cloud:

$$\mathcal{M}_{t+1} = \mathcal{R}(\mathbf{x}_{0:t}, \boldsymbol{\delta}_{0:t}, \boldsymbol{\alpha}_{0:t})$$

6: **end for**
7: **return** The initial $360°$ panoramic world view $x_0$ and a sequence of generated panoramic states $\mathbf{x}_{1:T}$, which together represent one explorable generative world, denoted as $\mathbf{x}_{0:T}$.

---

To provide spatial cues for the next generation step, we render target images of the 3D scene $\mathcal{M}_t$ by reprojecting the colored point cloud into the desired views using GPU-accelerated rasterization. Each 3D point is projected via a perspective transformation $\mathcal{T}$, defined by camera pose (translations $\boldsymbol{\delta}_t$ and rotations $\boldsymbol{\alpha}_t$). Since intrinsics are fixed across views in our setting, we omit them for simplicity. The renderer computes 2D coordinates, resolves visibility via depth buffering, and assigns pixel values from point colors. The rendered images are denoted as:

$$\mathbf{r}_t = \mathcal{T}(\mathcal{M}_t) \tag{1}$$

The rendered cubemap faces are converted to equirectangular format to match the diffusion model's input. Given the latest frame $x_{t-1}^S$ (with $x_0^S := x_0$), target poses $\boldsymbol{\delta}_t, \boldsymbol{\alpha}_t$, and 3D reprojections $\mathbf{r}_t$, we sample the next video clip $\mathbf{x}_t$ using the conditional video diffusion model:

$$\mathbf{x}_t \sim p_\theta(\mathbf{x} \mid x_{t-1}^S, \boldsymbol{\delta}_t, \boldsymbol{\alpha}_t, \mathbf{r}_t) \tag{2}$$

Once a new video clip is generated, it is integrated into the evolving 3D scene using a reconstruction function $\mathcal{R}$, which updates the memory map based on all past observations:

$$\mathcal{M}_{t+1} = \mathcal{R}(\mathbf{x}_{0:t}, \boldsymbol{\delta}_{0:t}, \boldsymbol{\alpha}_{0:t}) \tag{3}$$

By repeating this procedure over $T$ steps, we obtain a sequence of panoramic video clips $\mathbf{x}_{1:T}$, forming an explorable world, denoted as $\mathbf{x}_{0:T}$, grounded in an initial panoramic observation.

## 2.3 Camera Pose Embedding and Conditioning Mechanism

**Pose embedding.** To incorporate camera control, we introduce spherecal Plücker embeddings to encode the position and orientation of the target views in panoramic setting. For a perspective camera, the Plücker coordinates can be computed as

$$\boldsymbol{\varphi}_t = [\mathbf{d}, \mathbf{c}_t \times \mathbf{d}] \in \mathbb{R}^{(S+1) \times 6} \tag{4}$$

where $\mathbf{d}$ is the unit ray direction of each pixel, and $\mathbf{c}_t$ is $S + 1$ camera center locations at step $t$. In our approach, we concatenate the Plücker embeddings with the latent features, maintaining the same spatial dimensions. Unlike the conventional formulation that uses rays on a planar image grid, we compute the unit ray directions from the center of the panoramic sphere to its surface. This results in

a spherical ray field, which we convert into an equirectangular image, analogous to the transformation applied to the panoramic input. This representation encodes view-dependent spatial information more accurately for panoramic scenes, thereby enhancing the pose-conditioned video generation.

**Conditions of the diffusion model.** To generate a new video clip $\mathbf{x}_t$, the video diffusion model is conditioned on three key inputs: (a) the final frame $x_{t-1}^S$ from the previous video clip, following the temporal conditioning strategy of (Lu et al., 2025); (b) the rendered reprojections $\mathbf{r}_t$ of the reconstructed 3D world $\mathcal{M}_t$ from the target views defined by $(\boldsymbol{\delta}_t, \boldsymbol{\alpha}_t)$; (c) the spherical plücker embedding $E_t$ for the target view poses $(\boldsymbol{\delta}_t, \boldsymbol{\alpha}_t)$. The conditioning signals are concatenated with the noisy latent along the channel dimension and passed to the denoising network, while $x_{t-1}^S$ is embedded with a frozen CLIP image encoder (Radford et al., 2021) and injected at multiple layers via cross-attention to provide semantic guidance. To enhance generalization and enable flexible inference-time control, we randomly drop individual conditioning signals during training.

### 2.4 Evolving Generation and 3D Memory Update

We generate long panoramic videos by iteratively extending short clips while maintaining a 3D memory of the explored scene. Starting from an initial frame $x_0$, each video segment $\mathbf{x}_t$ is generated conditioned on the last frame of the previous segment, camera pose $(\boldsymbol{\delta}_t, \boldsymbol{\alpha}_t)$, and renderd reprojections $\mathbf{r}_t$ from the current 3D memory $\mathcal{M}_t$. After each generation step, the new clip $\mathbf{x}_t$ is used to update the 3D memory via a reconstruction function. Repeating this process produces a spatially consistent 3D scene and a long video trajectory grounded in the initial panoramic observation.

## 3 Experiments

We evaluate EvoWorld across multiple settings and tasks, comparing it with existing baselines. Section 3.1 covers implementation details. Section 3.2 reports video generation results across four datasets, analyzing spatial and temporal consistency. Section 3.3 presents ablations on camera pose conditioning and 3D memory. Section 3.4 evaluates downstream tasks. Section 3.5 analyzes inference speed, and Section 3.6 discusses the advantages of panoramic videos and limitations of the method.

### 3.1 Experimental Setup

**Dataset.** We introduce *Spatial360*, a large-scale dataset of high-quality panoramic videos across four domains: synthetic Unity and Unreal Engine 5 (UE5) scenes, indoor environments from HM3D (Ramakrishnan et al., 2021) and Matterport3D (Chang et al., 2017) via Habitat (Savva et al., 2019), and real-world outdoor captures. Each video is paired with ground-truth camera poses. Unlike prior panoramic datasets with unstable motion, low resolution, and inconsistent frame rates, *Spatial360* provides clean, stable, and well-annotated sequences tailored for controllable generation. It comprises 7,200 Unity, 10,000 UE5, 34,000 Habitat, and 7,200 real-world clips, each spanning 49–97 frames. This dataset establishes a strong benchmark for panoramic video generation and 3D-aware scene understanding. Further details are in Appendix A.2.

**Baselines.** We adopt Stable Video Diffusion (SVD) (Blattmann et al., 2023a) as our backbone, following GenEx (Lu et al., 2025), generating 25-frame clips at $1024 \times 576$ resolution. To improve memory efficiency, we omit the spherical-consistency loss used in prior works, applying this simplification to both GenEx and our model. For comparison, we fine-tune several image-to-video (I2V) diffusion models, LTX-Video-2B (HaCohen et al., 2024a), Wan-2.1-1.3B (Wan et al., 2025), CogVideoX-1.5-5B (Yang et al., 2024c), on our dataset. We also establish strong baselines by integrating CogVideoX-1.5-5B with our SpherePlücker representation and by fine-tuning ViewCrafter (Yu et al., 2024c), which are diffusion models that support camera-trajectory control.

**Implementation Details.** We extend VGGT (Wang et al., 2025) beyond reconstruction by aligning the estimated camera poses with ground truth in convention, scale, and rotation. The aligned point clouds are then reprojected into target views using a GPU rasterizer (Zhou et al., 2018). To ensure efficiency, we adopt **locality-aware retrieve-and-reproject strategy**, selecting a capped number of nearby frames to maintain constant memory usage over long trajectories. A **high-confidence threshold** further suppresses artifacts and filters dynamic elements. Training uses a batch size of 4

with gradient accumulation 4, a learning rate of $1 \times 10^{-5}$, and a cosine schedule with 500 warm-up steps, completing in 24h on 4 H100 GPUs. Additional details are provided in Appendix A.5.

## 3.2 Video Generation Quality

| Model | 2D Metrics | | | | | 3D Metrics | |
|---|---|---|---|---|---|---|---|
| | FVD ↓ | LMSE ↓ | LPIPS ↓ | PSNR ↑ | SSIM ↑ | MEt3R ↓ | AUC@30 ↑ |
| LTX-Video-2B (HaCohen et al., 2024b) | 317.54 | 0.137 | 0.456 | 17.18 | 0.733 | 0.1146 | 0.6431 |
| Wan-2.1 (Wan et al., 2025) | 228.59 | 0.115 | 0.467 | 16.00 | 0.697 | 0.1137 | 0.4192 |
| CogVideoX-1.5-5B (Yang et al., 2024c) | 205.57 | 0.109 | 0.430 | 15.78 | 0.716 | 0.1076 | 0.6527 |
| CogVideoX-1.5-5B + Our SpherePlücker | 148.94 | 0.071 | 0.259 | 19.02 | 0.778 | 0.1066 | 0.8125 |
| ViewCrafter (Yu et al., 2024c) | 168.27 | 0.097 | 0.353 | 18.04 | 0.758 | 0.0985 | 0.7273 |
| GenEx (Lu et al., 2025) | 199.76 | 0.113 | 0.400 | 17.11 | 0.743 | 0.1117 | 0.6408 |
| **EvoWorld** | **106.81** | **0.065** | **0.167** | **22.03** | **0.826** | **0.0954** | **0.8846** |

Table 1: Quantitative results on 25-frame panoramic video generation on Unity dataset. We report 2D metrics (FVD, MSE, LPIPS, PSNR, SSIM) and 3D metrics (MEt3R (Asim et al., 2025) and AUC@30 (Wang et al., 2025)). Arrows indicate whether lower (↓) or higher (↑) is better. All models are fine-tuned on the same panoramic dataset. EvoWorld achieves the best across all metrics.

We evaluate video generation quality using 2D metrics: FVD (Unterthiner et al., 2019), latent MSE (LMSE) (Lu et al., 2025), LPIPS (Zhang et al., 2018), PSNR (Horé & Ziou, 2010), and SSIM (Wang et al., 2004), and 3D metrics: spatial consistency (MEt3R (Asim et al., 2025)) and camera relocation accuracy (AUC@30 (Wang et al., 2025), which measures how well camera poses estimated from generated videos align with ground truth). Full metric definitions are provided in Appendix A.8.

**Single-clip generation quality.** Table 1 reports 25-frame single-clip results (Ren et al., 2025). Pretrained image-to-video models (COSMOS (Agarwal et al., 2025), Wan 2.1 (Wan et al., 2025), CogVideoX (Yang et al., 2024c)) generalize poorly to panoramic data, yielding high FVD (e.g., CogVideoX 820.13). Fine-tuning CogVideoX on *Spatial360* reduces errors; GenEx (Lu et al., 2025) improves temporal fidelity but suffers from cross-view inconsistency (MEt3R 0.1117, AUC@30 0.6408). ViewCrafter (Yu et al., 2024c) improves consistency (MEt3R 0.0985, AUC@30 0.7273). Our SpherePl"ucker extension further boosts reasoning (AUC@30 0.8125). EvoWorld achieves the best overall performance, cutting FVD to 106.81, LPIPS to 0.167, MEt3R to 0.0954, and raising AUC@30 to 0.8846, highlighting enhanced perceptual and spatial consistency.

**Recursive clip generation from a single frame.** We evaluate long-horizon performance by recursively generating three 25-frame segments (73 frames total) from a single panoramic input, where each segment conditions on the last frame of the previous one. This setup simulates extended navigation from minimal context. As shown in Table 2, EvoWorld consistently outperforms GenEx across Unity, UE5, indoor, and real-world 360° videos. On Unity, EvoWorld reduces FVD (491.71→442.79), improves PSNR (14.56→15.50), and lowers LPIPS (0.517→0.494). On UE5, it achieves a large FVD drop (516.85→431.37) with substantial PSNR (12.04→16.63) and SSIM (0.420→0.500) gains, showing robustness in complex outdoor scenes. Indoors, both methods struggle with clutter and occlusion, but EvoWorld still improves PSNR (12.99 vs. 12.16) and SSIM (0.559 vs. 0.545). On real-world data, despite sensor noise and dynamics, EvoWorld again improves across metrics, e.g., reducing FVD (988.62→908.10) and LPIPS (0.606→0.583). Results show that incorporating 3D memory substantially improves long-horizon coherence and perceptual quality across diverse and challenging environments. Qualitative comparisons are in Figure 3, with more results in Appendix A.1.2.

**Loop Consistency.** We evaluate whether models preserve spatial fidelity when completing closed trajectories using the Loop LMSE metric (Lu et al., 2025), which measures the latent MSE between the initial ground-truth frame and the final generated frame after a loop. Lower values indicate better global consistency and less drift. As shown in Table 2, EvoWorld achieves lower Loop LMSE than GenEx on most datasets, e.g., 0.192→0.187 on Unity and 0.199→0.151 on UE5, demonstrating reduced long-term drift. On Habitat, EvoWorld is slightly higher (0.235 vs. 0.228), likely due to tighter layouts and occlusions that challenge viewpoint-aligned memory. Overall, results confirm that 3D memory enhances cycle consistency in long-horizon video generation.

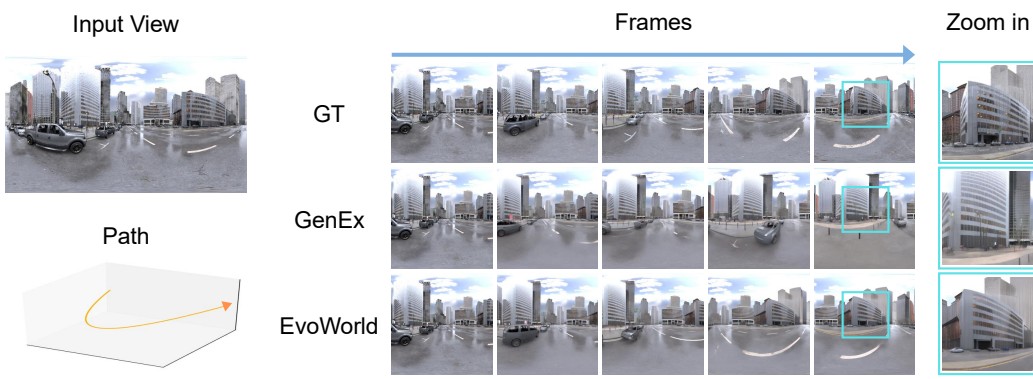

Figure 3: Qualitative comparison of long-horizon video generation. EvoWorld produces more spatially consistent and geometrically coherent results than GenEx. In this example, EvoWorld accurately follows the conditional path and preserves building structure, while GenEx struggles to maintain layout consistency due to the absence of 3D memory and precise camera control.

| | | FVD↓ | Loop LMSE↓ | SSIM↑ | PSNR↑ | LPIPS↓ | LMSE↓ |
|---|---|---|---|---|---|---|---|
| Unity | GenEx | 491.71 | 0.192 | 0.714 | 14.558 | 0.517 | 0.184 |
| | EvoWorld | **442.79** | **0.187** | **0.730** | **15.495** | **0.494** | **0.173** |
| UE5 | GenEx | 516.85 | 0.199 | 0.420 | 12.042 | 0.594 | 0.192 |
| | EvoWorld | **431.37** | **0.151** | **0.500** | **16.630** | **0.416** | **0.148** |
| Indoor | GenEx | 649.29 | **0.228** | 0.545 | 12.164 | 0.665 | **0.218** |
| | EvoWorld | **570.44** | 0.235 | **0.559** | **12.990** | **0.624** | **0.218** |
| Real-World | GenEx | 988.62 | 0.205 | 0.383 | 12.743 | 0.606 | 0.191 |
| | EvoWorld | **908.10** | **0.197** | **0.396** | **13.439** | **0.583** | **0.183** |

Table 2: **Quantitative comparisons on long-horizon video generation.** We evaluate recursive generation of three clips from a single panoramic frame, with each segment conditioned on the last frame of the previous one. Overall, EvoWorld outperforms GenEx across metrics on four domains (Unity, UE5, Indoor, and Real-World), showing clear gains in fidelity and perceptual quality.

## 3.3 ABLATION STUDY

| Model | 2D Metrics | | | | | 3D Metrics | |
|---|---|---|---|---|---|---|---|
| | FVD ↓ | LMSE ↓ | LPIPS ↓ | PSNR ↑ | SSIM ↑ | MEt3R ↓ | AUC@30 ↑ |
| GenEx (Lu et al., 2025) | 199.76 | 0.113 | 0.400 | 17.11 | 0.743 | 0.1117 | 0.6408 |
| Baseline + GCD (Van Hoorick et al., 2024) | 178.75 | 0.109 | 0.389 | 17.33 | 0.745 | 0.1127 | 0.6700 |
| Baseline + SpherePlücker | 162.96 | 0.090 | 0.275 | 19.54 | 0.785 | 0.1070 | 0.7958 |
| **EvoWorld** | **106.81** | **0.065** | **0.167** | **22.03** | **0.826** | **0.0954** | **0.8846** |

Table 3: Ablation study on the Unity dataset evaluating the impact of 3D memory and camera pose representations. "GCD": Generative Camera Dolly (Van Hoorick et al., 2024); "SpherePlücker": our Spherical Plücker representation for panoramic videos. Combining SpherePlücker embeddings with 3D memory (EvoWorld) yields the best performance across all metrics.

Table 3 reports an ablation on camera pose representation and 3D memory in the curved Unity setting. Starting from GenEx (Lu et al., 2025), using Generative Camera Dolly (GCD) (Van Hoorick et al., 2024) as camera condition improves both 2D and 3D metrics, while our Spherical Plücker embeddings yield larger gains (FVD 162.96, AUC@30 0.7958), highlighting stronger spatial conditioning. Adding our 3D memory module on top of SpherePlücker (EvoWorld) achieves the best results (FVD 106.81, SSIM 0.826, MEt3R 0.0954, AUC@30 0.8846). Overall, 3D memory and Spherical Plücker proves critical for spatial coherence, together enabling more consistent panoramic video generation.

## 3.4 EVALUATION ON DOWNSTREAM TASKS.

To evaluate the spatial reasoning and utility of our generated panoramic videos, we assess EvoWorld on two downstream tasks: target reaching and spatially-aware frame retrieval (Table 4). Both tasks require precise spatial consistency and are evaluated using GPT-4o (Hurst et al., 2024) as a vision-language model (VLM) to interpret the generated content. The detailed description and examples of the downstream tasks are in Appendix A.6.

| Algorithm | Target reaching | Frame retrieval | Average |
|---|---|---|---|
| GPT-4o | 45.0% | N/A | N/A |
| GPT-4o + GenEx | 83.5% | 50.5% | 67.0% |
| GPT-4o + EvoWorld | **93.3%** | **68.8%** | **81.1** % |

Table 4: Downstream task performance evaluating spatial consistency with GPT-4o (Hurst et al., 2024) as the evaluator. We report target-reaching accuracy and spatial frame-retrieval accuracy. EvoWorld outperforms GenEx on both tasks, demonstrating stronger spatial grounding and controllability in panoramic world generation.

**Target reaching.** This task measures whether synthesized views support navigation toward a specified target. Given a source view, a target view, and four candidate navigation directions, only one path correctly leads to the target. GPT-4o alone achieves 45.0% accuracy, which improves to 83.5% with GenEx and further to 93.3% with EvoWorld.

**Spatially-aware frame retrieval.** We test whether generated frames align with their true spatial location. From a 3-clip (73-frame) video, a single frame is sampled and matched against four ground-truth candidates: one correct and three distractors offset by 2/4/6m. GenEx achieves 50.5% accuracy, while EvoWorld reaches 68.8%, indicating stronger spatial grounding and global layout preservation.

**Generated videos for 3D reconstruction.** We further examine whether generated videos can benefit 3D reconstruction when ground-truth inputs are sparse. As shown in Figure 4, using only four real images yields incomplete reconstructions with holes and poor coverage. Augmenting with GenEx-generated frames improves coverage but adds noise from spatial inconsistency. In contrast, adding EvoWorld-generated frames produces more complete, coherent, and artifact-free reconstructions, highlighting their geometric value for downstream tasks such as SLAM and scene reconstruction.

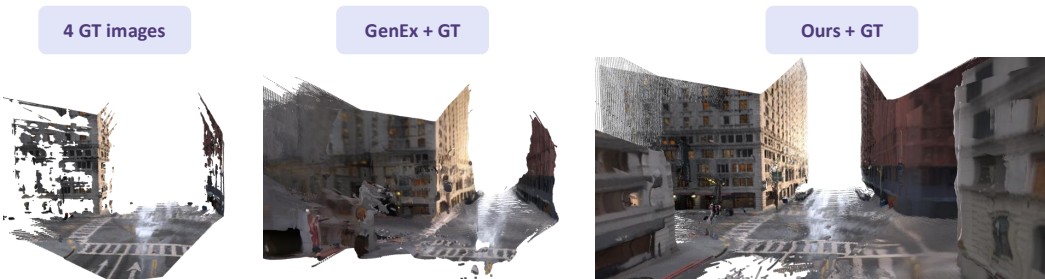

Figure 4: Qualitative comparison of 3D reconstructions using four ground truth (GT) images alone, GT with GenEx-generated frames, and with EvoWorld (in the same scale). GT-only reconstructions are incomplete with holes; adding GenEx frames improves coverage but introduces noise. EvoWorld yields more complete and cleaner reconstructions, demonstrating better spatial consistency.

Overall, these results demonstrate that EvoWorld produces videos with stronger spatial consistency and controllability. High accuracy in target reaching shows precise directional reasoning, improved retrieval indicates global consistency with real geometry, and enhanced reconstructions confirm meaningful 3D grounding. Together, they validate the potential of EvoWorld for embodied AI.

## 3.5 EFFICIENCY ANALYSIS

Table S3 shows FPS performance: CogVideoX-1.5 runs slowest (0.12), while GenEx and EvoWorld reach 0.33 and 0.32; memory updates alone achieve 4.20 FPS, showing efficiency relative to generation, with combined updates only slightly reduced to 0.30. Since our framework can swap in faster backbones easily, future advances in generation or reconstruction will directly improve FPS.

## 3.6 DISCUSSION AND LIMITATIONS

**Advantages of panoramic videos.** Panoramic videos provide richer spatial coverage and enable impactful applications, such as embodied AI (Koh et al., 2021; Wang et al., 2024d), Virtual/Augmented

Reality (Koh et al., 2022; Broxton et al., 2020), and spatial video editing (SIG, 2025). Panoramic video generation is an emerging research direction with distinct challenges (Luo et al., 2025; Xia et al., 2025; Tencent, 2025). Our framework provides a new dataset and demonstrates robustness across both synthetic and real-world settings, laying a foundation for scalable panoramic world modeling.

**Limitations.** Although our results are promising, we have not yet explored very long-horizon settings, as current open-source generators typically max out at a few hundred frames. Longer-horizon models can be integrated seamlessly once available. Our approach also inherits the quality of the reconstruction backbone, so advances in reconstruction will improve the performance. Finally, expanding evaluation metrics and scaling the dataset would further strengthen the benchmark.

## 4 RELATED WORK

**Generative video modeling.** Diffusion models (DMs) (Sohl-Dickstein et al., 2015; Ho et al., 2020) have achieved impressive image synthesis, with latent diffusion models (LDMs) (Rombach et al., 2022) enabling efficient high-resolution generation in latent space. Extending this to video, recent approaches (Blattmann et al., 2023c;b; Wang et al., 2023a; Blattmann et al., 2023a; Song et al., 2025; Guo et al., 2023; Luo et al., 2023) employ VAEs to encode frames and denoise in latent space. Controllable synthesis has been explored via text (Rombach et al., 2022; OpenAI, 2024) and other conditional inputs (Zhang et al., 2023; Sudhakar et al., 2024). Despite progress, state-of-the-art video generators remain largely ungrounded in physical environments, limiting their use as world models that support action-conditioned reasoning.

**Generative world models.** World models aim to predict future states for planning and control (Ha & Schmidhuber, 2018; LeCun, 2022), though early efforts were limited to simple agents without physical grounding. More recent works leverage generative vision (OpenAI, 2024; Kondratyuk et al., 2024) and video in-context learning (Bai et al., 2024; Zhang et al., 2024) for real-world decision-making (Yang et al., 2024b). Domain-specific systems for driving (Hu et al., 2023; Wang et al., 2023b; 2024c; Gao et al., 2024a;b) and instructional video generation (Du et al., 2024a; Yang et al., 2024a; Wang et al., 2024a; Bu et al., 2024; Souček et al., 2024; Du et al., 2024b) demonstrate utility but lack generality. Interactive generative world models have recently emerged for exploration and navigation (Lu et al., 2025; Bar et al., 2025; Xiao et al., 2025), while concurrent efforts introduce persistent spatial memory into video world models (Wu et al., 2025; Li et al., 2025; Zhou et al., 2025). These works highlight the importance of memory for maintaining temporal and spatial coherence, but rely on implicit representations or do not address panoramic settings. In contrast, our framework introduces **explicit 3D memory** that evolves alongside video generation, providing stronger geometric grounding for long-horizon panoramic world modeling.

**3D reconstruction.** Traditional 3D reconstruction methods, such as Structure from Motion (SfM) and Multi-View Stereo (MVS), can recover 3D models from 2D image collections via triangulation. NeRF (Mildenhall et al., 2021) and Gaussian Splatting (Kerbl et al., 2023) can deliver photo-realistic rendering quality. However, these methods often struggle when extrapolating to faraway viewpoints. Incorporating temporal prior from videos has opened new avenues for 3D reconstruction. ReconX (Liu et al., 2024) uses a video diffusion model to help improve reconstruction quality. WonderJourney (Yu et al., 2024b) and WonderWorld (Yu et al., 2024a) have used graphic primitive Gaussian surfels, preserving 3-D detail yet supporting only narrow camera paths (linear or rotational) and yielding context gaps that cause local inconsistencies. The recent feed-forward systems, such as VGGT (Wang et al., 2025), perform accurate, real-time geometry recovery. Yet, 3D reconstruction from generated video using feed-forward methods has not been thoroughly explored.

## 5 CONCLUSION

We presented a 3D memory-augmented framework for panoramic world generation, together with *Spatial360*, a new large-scale dataset for this task. By jointly evolving video generation and updating 3D scenes, our method reduces spatial drift and delivers more consistent long-horizon synthesis. Experiments across synthetic, indoor, and real-world domains demonstrate significant gains in both fidelity and coherence. Collectively, the framework and dataset establish a foundation for future research on scalable, physically grounded world models with explicit 3D memory.

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

# A APPENDIX

## A.1 EXTENDED EXPERIMENTS

### A.1.1 EXTENDED QUANTITATIVE RESULTS

In this section, we present more results on quantitative comparison of single-clip(25-frame) panoramic video generation on the Unity dataset(extension of Table 1). Compared with the main paper, we add more baselines here (LTX-Video (HaCohen et al., 2024b) and Wan-2.1 (Wan et al., 2025) with and without finetuning on our *Spatial360*).

| Model | Polyline Paths | | | | | Curved Paths | | | | |
|---|---|---|---|---|---|---|---|---|---|---|
| | FVD ↓ | MSE ↓ | LPIPS ↓ | PSNR ↑ | SSIM ↑ | FVD ↓ | MSE ↓ | LPIPS ↓ | PSNR ↑ | SSIM ↑ |
| → *Direct test* | | | | | | | | | | |
| SVD (Blattmann et al., 2023a) | 646.72 | 0.169 | 0.503 | 14.63 | 0.691 | 612.25 | 0.164 | 0.488 | 14.93 | 0.694 |
| CogVideoX-1.5 (Yang et al., 2024c) | 800.29 | 0.159 | 0.469 | 15.52 | 0.686 | 820.13 | 0.163 | 0.486 | 15.28 | 0.679 |
| COSMOS (Agarwal et al., 2025) | 1058.40 | 0.195 | 0.582 | 14.78 | 0.638 | 1035.93 | 0.196 | 0.586 | 14.76 | 0.639 |
| Wan-2.1 (Wan et al., 2025) | 595.57 | 0.138 | 0.448 | 16.62 | 0.692 | 506.44 | 0.139 | 0.513 | 15.45 | 0.668 |
| → *Tune on Spatial360* | | | | | | | | | | |
| CogVideoX-1.5 (Yang et al., 2024c) | 120.08 | 0.055 | 0.163 | 21.67 | 0.834 | 205.57 | 0.109 | 0.430 | 15.78 | 0.716 |
| Wan-2.1 (Wan et al., 2025) | 91.32 | 0.061 | 0.245 | 19.96 | 0.775 | 228.59 | 0.115 | 0.467 | 16.00 | 0.697 |
| GenEx (Lu et al., 2025) | 70.03 | 0.046 | 0.123 | 24.19 | 0.865 | 199.76 | 0.113 | 0.400 | 17.11 | 0.743 |
| **EvoWorld** | **61.19** | **0.041** | **0.108** | **24.36** | **0.869** | **106.81** | **0.065** | **0.167** | **22.03** | **0.826** |

Table S1: Quantitative comparison of single-clip (25-frame) panoramic video generation on the Unity dataset under two types of camera trajectories: polyline paths and curved paths. Models are evaluated using FVD, MSE, LPIPS, PSNR, and SSIM, where ↓ indicates lower is better and ↑ indicates higher is better. Our method, EvoWorld, achieves the best performance across all metrics in both trajectory settings.

### A.1.2 QUALITATIVE RESULTS

In this section, we present qualitative results across the four subsets of the *Spatial360* dataset: Unity, Unreal Engine (UE), Indoor, and Real-World environments. They are shown in Figure S1, Figure S2, Figure S3, Figure S4, respectively. These visualizations highlight the spatial and temporal consistency of generated panoramic videos under diverse conditions.

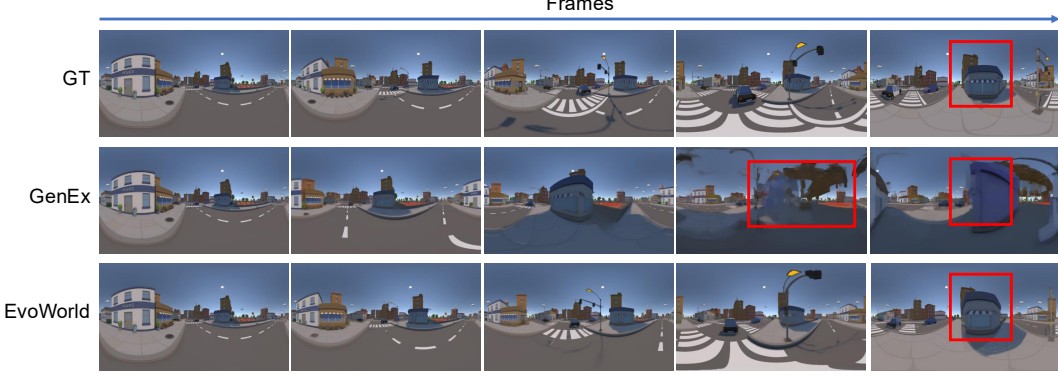

Figure S1: Qualitative results of panoramic video generation in the **Unity** environment. Five key frames are shown from a longer video, with intermediate frames omitted for brevity. GenEx exhibits spatial drift and structural artifacts, while our method (EvoWorld) maintains spatial consistency throughout.

## A.2 SPATIAL360 DETAILS

*Spatial360* is a high-quality panoramic video dataset spanning four domains. All sequences are rendered or captured at a resolution of $2000 \times 1000$ pixels and paired with ground-truth camera

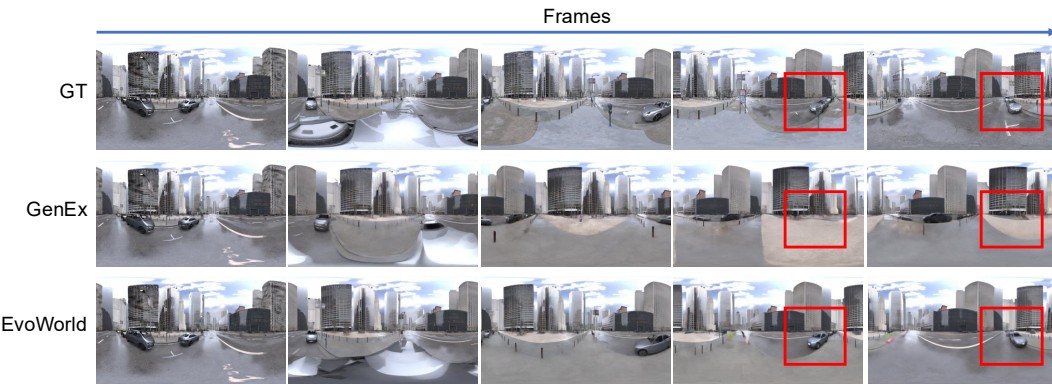

Figure S2: Qualitative results of panoramic video generation in the **Unreal Engine 5 (UE5)** environment. Five key frames are shown from a longer video, with intermediate frames omitted for brevity. GenEx exhibits spatial drift and missing object, while our method (EvoWorld) maintains spatial consistency throughout.

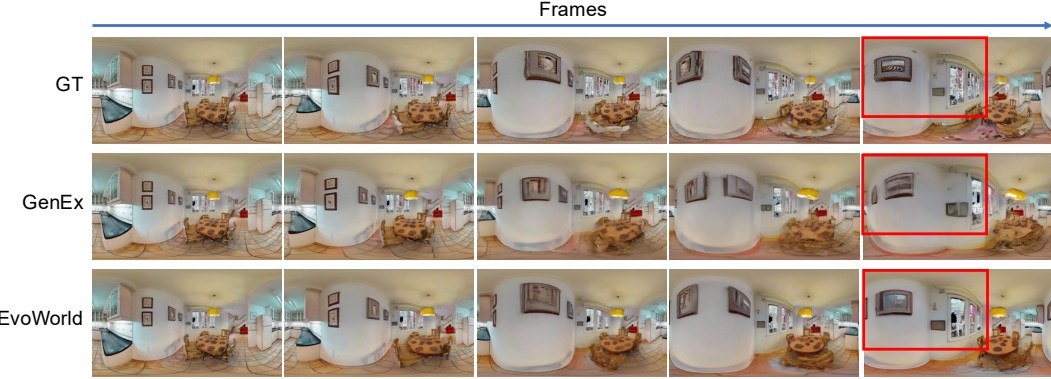

Figure S3: Qualitative results of panoramic video generation in the **indoor** environment. Five key frames are shown, with intermediate frames omitted for brevity. GenEx exhibits inconsistent object shapes and numbers, while our method (EvoWorld) maintains spatial consistency throughout.

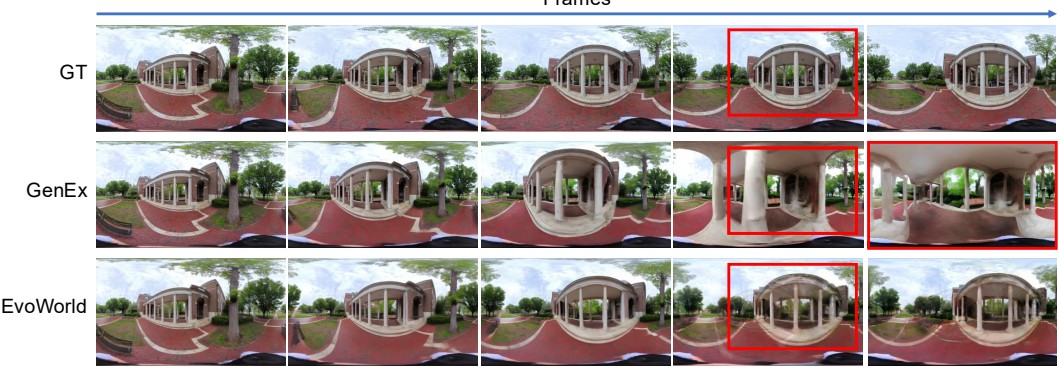

Figure S4: Qualitative results of panoramic video generation in the **real-world** environment. Five key frames are shown from a longer video, with intermediate frames omitted for brevity. GenEx exhibits spatial drift and structural artifacts, while our method (EvoWorld) maintains spatial consistency throughout.

poses (3D positions and quaternion orientations). Each trajectory consists of 49–97 frames, which are further segmented into 2–4 overlapping clips of 25 frames. Below we detail each data source.

| Environment | # Videos | # Frames | Distance (m) | Time (s) | Navigation Trajectory |
|---|---|---|---|---|---|
| Unity | 7,200+ | 680,000+ | 260,000+ | 97,000+ | Polyline / Curve |
| UE5 | 10,000+ | 1,000,000+ | 400,000+ | 140,000+ | Curve |
| Indoor | 34,000+ | 540,000+ | 210,000+ | 77,000+ | Step-wise Forward or 22.5° Rotation |
| Real-world | 7,200+ | 540,000+ | 5,400+ | 9,000+ | Natural Human Walk (Curve) |

Table S2: Statistics of the four subsets in the *Spatial360* dataset.

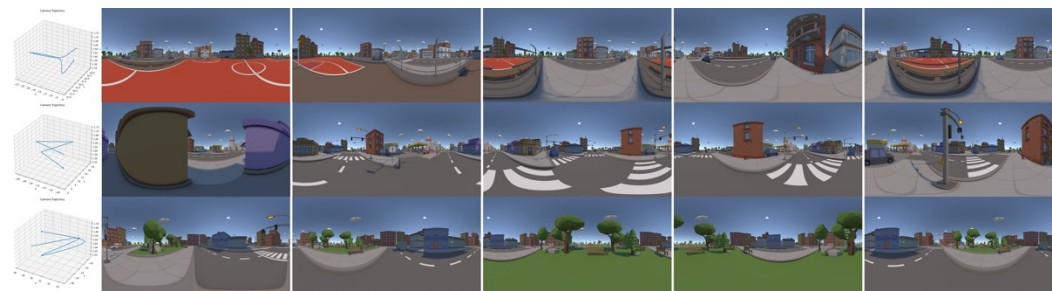

Figure S5: **Examples from Unity Dataset.** Each row is a case in the Unity dataset. The first column is camera trajectories visualized in a 3D coordinate system. The remaining five are evenly sampled five frames from the raw sequence.

**Unity (Synthetic).** This subset contains 3,600 clips rendered from Unity environments. Trajectories are generated with a step size of $0.4$ m and lengths of 20, 30, or 40 m. Two path types are included: polyline trajectories, which are looped with minimal start–end displacement, and curved trajectories obtained by Catmull–Rom spline smoothing. Ground-truth camera poses are provided for every frame, including both positions $(x, y, z)$ and orientations $(w, x, y, z)$. You can find three examples with a curved/polyline camera path in Fig. S5

**Unreal Engine 5 (Synthetic).** This subset comprises 10,000 clips rendered in UE5 environments. All sequences are generated using curve-based trajectories with a step size of $0.4$ m and lengths of 20, 30, or 40 m. Each frame is annotated with ground-truth camera poses including 3D positions and quaternion orientations(Fig. S6).

**Habitat (Indoor).** This subset includes 34,000 clips simulated in Habitat using reconstructions from HM3D (Ramakrishnan et al., 2021) and Matterport3D (Chang et al., 2017). Trajectories are defined as non-looped polylines with a step size of $0.4$ m and are constructed from random action sequences consisting of forward moves of $0.4$ m and rotations of $\pm22.5°$, which yield diverse and stochastic navigation patterns. Each frame is paired with ground-truth camera poses of positions and orientations(first three rows in Fig. S7).

**Real-World (Insta360).** This subset consists of 28,000 outdoor clips, totaling approximately two hours of footage captured at 5 FPS with a handheld Insta360 camera. Trajectories exhibit variable step

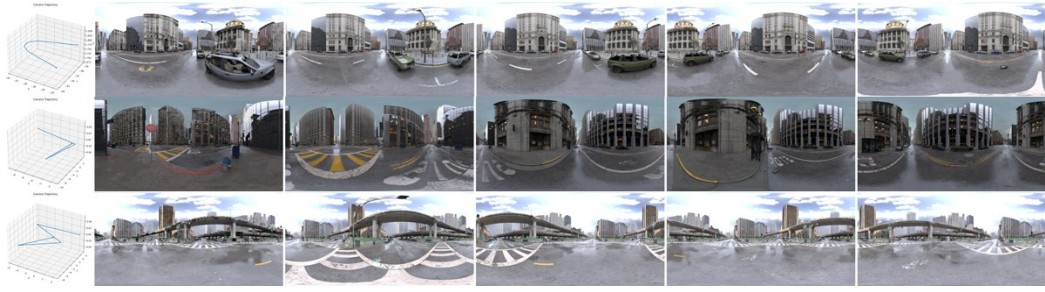

Figure S6: **Examples from UE5 Dataset.** Each row is a case in the UE5 dataset. The first column is camera trajectories visualized in a 3D coordinate system. The remaining five are evenly sampled five frames from the raw sequence.

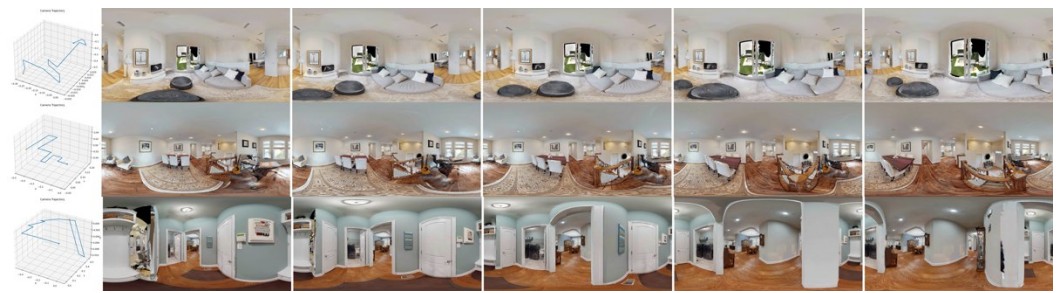

Figure S7: Examples from Indoor Dataset.

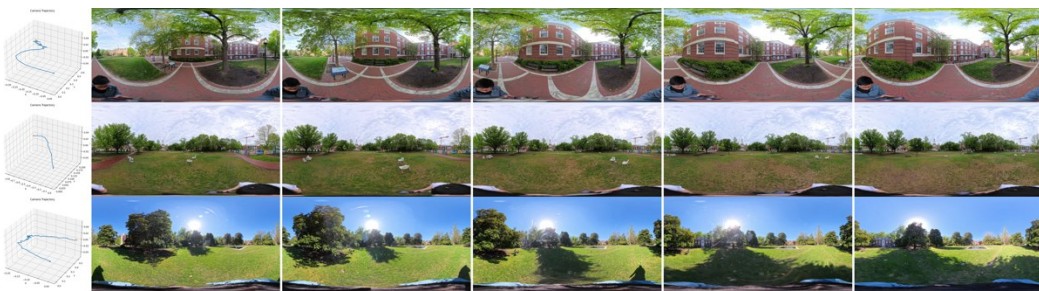

Figure S8: Examples from Real-world Dataset.

sizes and are typically organized as looped curves. Camera poses are estimated using DROID-SLAM, producing reliable 3D positions and quaternion orientations for every frame(Fig. S8).

In summary, *Spatial360* unifies synthetic, indoor, and real-world panoramic video data into a consistent format, enabling controlled evaluation of panoramic video generation and advancing research in 3D-aware scene understanding (see Table S2 for detailed statistics).

## A.3 DISCUSSIONS

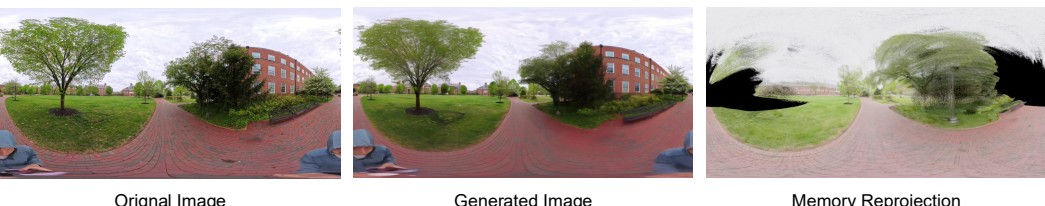

Figure S9: Comparison of (a) target image, (b) generated result, and (c) reprojection visualization. During memory construction, a relatively high threshold filters out the moving person and parts of the trees swaying in the wind. However, in the generation stage, the model still generates corresponding human and tree structures based on the input image elements.

## A.4 INFERENCE SPEED

See Tab. S3 for the listed speed comparison.

## A.5 ADDITIONAL IMPLEMENTATION DETAILS

We fine-tune models initialized from Stable Video Diffusion (SVD) separately on each domain: 30,000 steps for Unity, UE5, and real-world data (initialized from the Unity model), and 10,000 steps for indoor scenes. Training uses a batch size of 4 with gradient accumulation over 4 steps, a learning rate of $1 \times 10^{-5}$, cosine schedule, 500-step warm-up, and runs for 24 hours on 4 H100 GPUs. To

Table S3: Comparison of generation and memory update speed (FPS).

| Model | Function | FPS |
|---|---|---|
| CogVideoX-1.5 | Generation | 0.12 |
| GenEx | Generation | 0.33 |
| EvoWorld | Generation | 0.32 |
| EvoWorld | Mem update | 4.20 |
| EvoWorld | Generation + Mem update | 0.30 |

incorporate 3D memory, we use VGGT Wang et al. (2025) to reconstruct colored point clouds from prior frames. These reconstructions and their reprojections are precomputed and used as conditions. Since VGGT-estimated poses may differ from the conditional target view poses in scale and rotation, we apply alignment before reprojecting into target views.

To incorporate 3D memory, we extend VGGT (Wang et al., 2025) to reconstruct colored point clouds from prior frames, which are then reprojected to target views as conditioning inputs. Rendering is accelerated using a GPU-based rasterizer (Zhou et al., 2018). Since VGGT outputs camera poses in the world-to-camera (OpenCV) convention, while Open3D adopts the OpenGL convention, we first align the coordinate systems before reprojection. Moreover, because VGGT-estimated poses may differ in scale or rotation from target views, we perform pose alignment to ensure consistency.

To enable efficient memory construction and retrieval, we adopt a **locality-aware retrieve-and-reproject strategy**: at each step, only past frames whose camera poses are spatially close to the current location are used for VGGT-based 3D reconstruction, avoiding caching all frames. To maintain scalability over long trajectories, we cap the number of VGGT inputs below 100 frames. This design ensures constant memory usage and stable inference, independent of sequence length.

Finally, we apply a **high confidence threshold** during reconstruction to suppress artifacts and filter out dynamic elements, thereby enhancing the stability of the evolving 3D memory.

## A.6 DOWNSTREAM TASKS DETAIL

### A.6.1 TASK DESCRIPTION

**Target reaching.** This task is designed to assess whether a world model can facilitate accurate navigation toward a specified target view and improve spatial reasoning. In this task, the model is given a source image (view) and a target image (view), along with four candidate navigation paths. Each path is defined as a fixed-length ray extending from the source location with one of four angular offsets: -9°, -3°, 3°, or 9°. Only one of these paths terminates at the target view. The objective is to identify the candidate path that correctly aligns with the target image. We consider two evaluation settings. In the first setting, with a world model, each candidate path is used as input to the world model to generate the panoramic view at its endpoint. A vision-language model (GPT-4o) then compares the four generated views with the target image and selects the best match. In the second setting, without a world model, GPT-4o is directly prompted with the source image, the target image, and the task description, and must select the correct direction from the four angular options based solely on visual and spatial reasoning. This design allows us to test two hypotheses: (1) whether the ability of a world model to synthesize future views improves the spatial reasoning capacity of a vision-language model, and (2) whether the proposed world model produces more accurate future views than a baseline model.

**Spatially-aware frame retrieval.** Here, we test whether generated frames align with their true spatial location. From a 3-clip (73-frame) video, a single frame is sampled and matched against four ground-truth candidates: one correct and three distractors offset by 2m, 4m, and 6m. GenEx achieves 50.5% retrieval accuracy, whereas EvoWorld reaches 68.8%, showing that 3D memory substantially improves spatial grounding and global layout preservation.

### A.6.2 PROMPTS FOR DOWNSTREAM SPATIAL QA TASKS

In this section, we present our image-text prompts to the Large Multi-modal Model (LMM) and image examples for downstream tasks, including:

**(a) Target Reaching**. Image-text prompt: Figure S10. Image examples: Figure S11.

**(b) Spatially-aware Frame Retrieval**. Image-text prompt: Figure S12. Image examples: Figure S13.

---

## LMM Prompts

**Target Reaching**

**With World Model:** You are given four panoramic images (A, B, C, D) generated from different viewpoints and one real panoramic image. Your task is to compare the real image with the four generated ones and determine which generated image was taken from the same position and orientation as the real image. Focus on visual cues such as scene layout, object positions, building angles, road structure, and overall viewpoint. Answer with a single uppercase letter from {A, B, C, D} that best matches the real image.

For $letter in A B C D:

Image $letter is a generated panoramic image: <IMAGE TOKEN $LETTER>.

This is the real panoramic image. It corresponds to one of the generated images A, B, C, or D. Please identify which one: <IMAGE TOKEN REAL>.

**Without World Model:** You are given five panoramic images. The first image is captured at the central position, while the remaining four images (labeled A, B, C, D) are taken from positions on the edge of a circle centered at the first image's location. The central image is oriented at 0 degrees, and the four edge images are oriented at -9°, -3°, 3°, and 9°, respectively. Given a target viewing angle from the set {-9, -3, 3, 9}, your task is to identify which of the four labeled images was captured at that orientation. Answer with a single uppercase letter from {A, B, C, D}. Note: A B C D don't have to be in the order of -9°, -3°, 3°, and 9°.

For $letter in A B C D:

Image $letter is an edge image possibly facing any degree: <IMAGE TOKEN $LETTER>.

This is the initial panoramic image located at centroid: <IMAGE TOKEN INIT>.

Figure S10: Text-image prompt template for target reaching task

### A.7 PRELIMINARY: EQUIRECTANGULAR PANORAMIC IMAGES

An *equirectangular panorama* records the full $360°$ scene from a single viewpoint and flattens that spherical content onto a 2D grid via a simple mapping. We define the *Spherical Polar Coordinate*: $\mathcal{S}$: Each point is written as $(\phi, \theta, r) \in \mathcal{S}$, where longitude $\phi \in [-\pi, \pi)$, latitude $\theta \in [-\pi/2, \pi/2]$, and radius $r > 0$.

We define the *Pixel Coordinate System* $\mathcal{P}$: A pixel on the image plane is $(u, v) \in \mathcal{P}$ with $u \in [0, W-1]$ (horizontal) and $v \in [0, H-1]$ (vertical).

We define the *Spherical $\leftrightarrow$ Pixel Mapping*.

$$f_{\mathcal{S}\to\mathcal{P}}(\phi, \theta) = \left( \frac{W}{2\pi}(\phi + \pi), \ \frac{H}{\pi}\left(\frac{\pi}{2} - \theta\right) \right), \tag{5}$$

$$f_{\mathcal{P}\to\mathcal{S}}(u, v) = \left( \frac{2\pi u}{W} - \pi, \ \frac{\pi}{2} - \frac{\pi v}{H} \right). \tag{6}$$

These forward and inverse transforms cover the full sphere without gaps.

A panorama therefore bundles every viewing direction from one spot, giving us global context that keeps generated content aligned with its 3D surroundings.

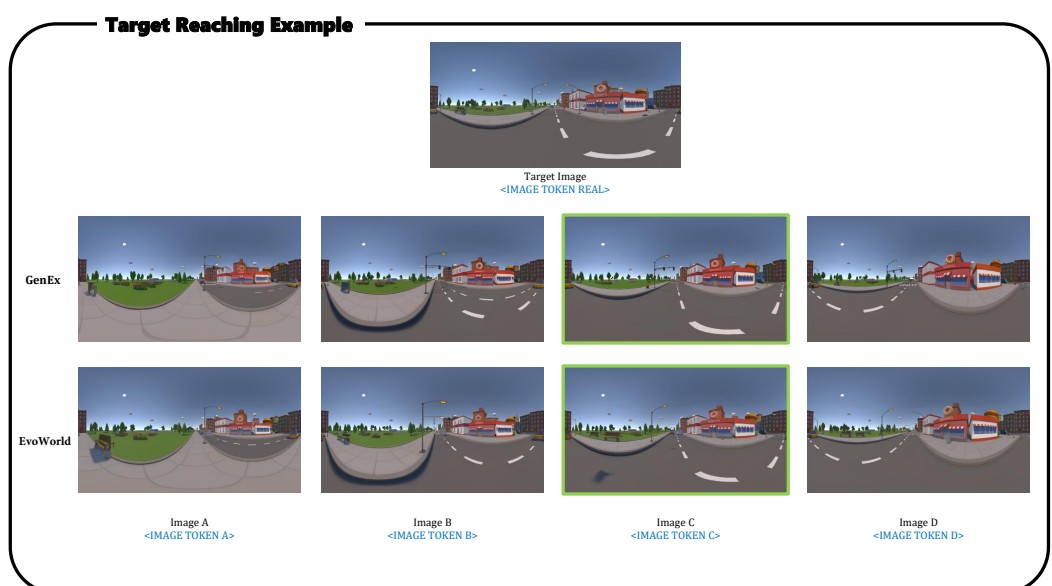

Figure S11: Image examples for target reaching task

Figure S12: Text-image prompt template for spatially-aware frame retrieval task

Because panoramas live on the sphere, we can rotate them to emulate head turns with no information loss, or unwrap them into six cube faces for standard 2D viewing (see Fig. S14).

We have the *Spherical Rotation* .

$$\mathcal{T}(u, v, \Delta\phi, \Delta\theta) = f_{\mathcal{S}\to\mathcal{P}}\big(\mathcal{R}(f_{\mathcal{P}\to\mathcal{S}}(u, v), \Delta\phi, \Delta\theta)\big), \tag{7}$$

with

$$\mathcal{R}(\phi, \theta, \Delta\phi, \Delta\theta) = \big((\phi + \Delta\phi) \bmod 2\pi, \ (\theta + \Delta\theta) \bmod \pi\big). \tag{8}$$

Default offsets are $\Delta\phi = \Delta\theta = 0$.

For Panorama→Cubemap transform: An equirectangular image can be split into the front, back, left, right, top, and bottom cube faces for easier display.

## A.8 EVALUATION DETAILS

We employ a suite of perceptual and pixel-wise metrics to evaluate the quality and consistency of generated videos:

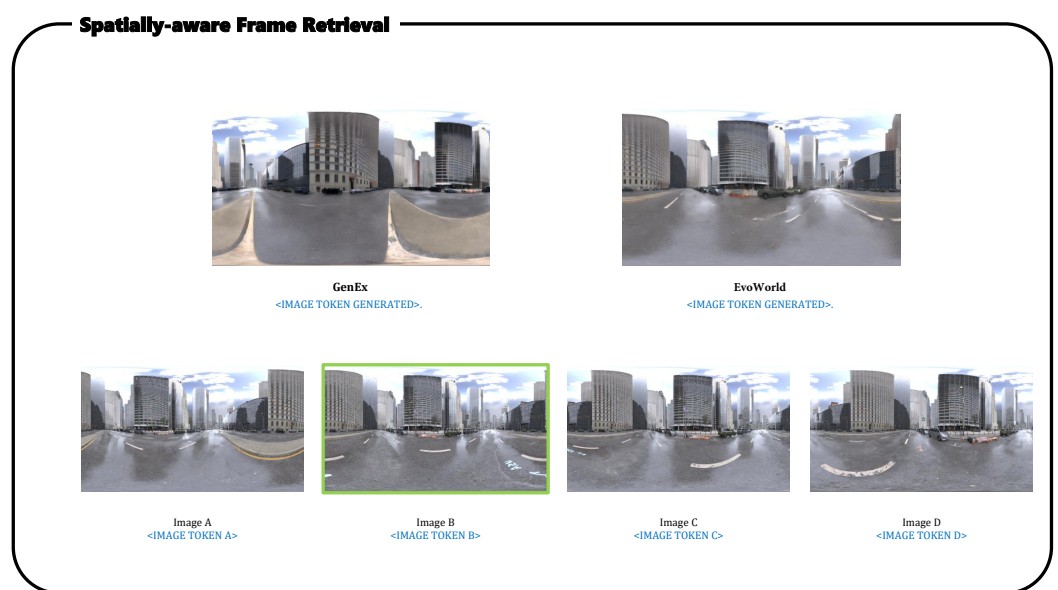

Figure S13: Image examples for spatially-aware frame retrieval task

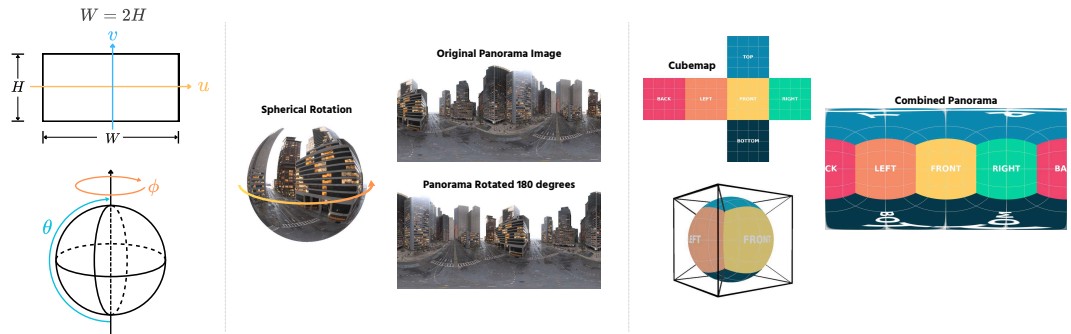

Figure S14: Left: Pixel coordinate and Spherical Polar coordinate systems; Middle: rotation in Spherical coordinates corresponds to rotation in 2D image; Right: expansion from panorama to cubemap or composition in reverse. The figures are borrowed from (Lu et al., 2025) for explanation.

**Structural Similarity Index (SSIM).** SSIM evaluates perceptual similarity between frames by comparing luminance, contrast, and structure. Given two image patches $x$ and $y$, SSIM is defined as:

$$\text{SSIM}(x, y) = \frac{(2\mu_x\mu_y + C_1)(2\sigma_{xy} + C_2)}{(\mu_x^2 + \mu_y^2 + C_1)(\sigma_x^2 + \sigma_y^2 + C_2)}$$

where $\mu_x, \mu_y$ are means, $\sigma_x^2, \sigma_y^2$ are variances, and $\sigma_{xy}$ is the covariance between $x$ and $y$. The constants $C_1$ and $C_2$ stabilize the division. SSIM ranges from 0 to 1, with higher values indicating more similarity.

**Peak Signal-to-Noise Ratio (PSNR).** PSNR measures reconstruction quality by comparing pixel-level fidelity. For a reference image $I$ and a generated image $\hat{I}$ with maximum pixel value $L$, it is defined as:

$$\text{PSNR}(I, \hat{I}) = 10 \cdot \log_{10}\left(\frac{L^2}{\text{MSE}(I, \hat{I})}\right)$$

where $\text{MSE}(I, \hat{I}) = \frac{1}{N}\sum_{i=1}^{N}(I_i - \hat{I}_i)^2$. Higher PSNR indicates better fidelity.

**Fréchet Video Distance (FVD).**  FVD extends the Fréchet Inception Distance (FID) to video. It compares the distribution of real and generated video embeddings extracted from a pretrained Inflated 3D ConvNet (I3D). Formally, for two multivariate Gaussians with means $\mu_r, \mu_g$ and covariances $\Sigma_r, \Sigma_g$ (from real and generated videos, respectively), the FVD is computed as:

$$\text{FVD} = \|\mu_r - \mu_g\|_2^2 + \text{Tr}\left(\Sigma_r + \Sigma_g - 2(\Sigma_r \Sigma_g)^{1/2}\right)$$

Lower values indicate greater similarity to real video distributions. We observed significant fluctuations in FVD scores when using different frame indices as the reference frame during evaluation. To mitigate this instability, we compute the average FVD over a number of frames range from 10 to last frame.

**Learned Perceptual Image Patch Similarity (LPIPS).**  LPIPS compares deep feature representations from a pretrained network (e.g., AlexNet or VGG) to assess perceptual similarity. For two images $x$ and $y$:

$$\text{LPIPS}(x, y) = \sum_l \frac{1}{H_l W_l} \sum_{h,w} \|w_l \odot (\phi_l(x)_{hw} - \phi_l(y)_{hw})\|_2^2$$

where $\phi_l$ denotes features from layer $l$ and $w_l$ are learned weights. Lower LPIPS indicates better perceptual similarity.

**Latent MSE.**  Latent MSE measures the Euclidean distance between latent embeddings of generated and ground-truth videos in a learned representation space. Let $z$ and $\hat{z}$ be the latent codes of the ground-truth and generated frames, respectively:

$$\text{Latent MSE} = \frac{1}{N} \sum_{i=1}^{N} \|z_i - \hat{z}_i\|_2^2$$

This metric reflects how well high-level video dynamics or appearance are preserved in the latent space, which can correlate with perceptual consistency.

**MEt3R.**  MEt3R (Multi-view Consistency Metric) (Asim et al., 2025) evaluates the geometric consistency of generated images or videos across different viewpoints, without requiring known camera poses or ground-truth depth. Given two generated views $I_1$ and $I_2$, MEt3R operates in four steps: (i) dense 3D point maps $X_1, X_2$ are reconstructed via MASt3R (Leroy et al., 2024) or DUSt3R (Wang et al., 2024b) in a shared coordinate frame; (ii) deep feature maps $F_1, F_2$ are extracted from $I_1, I_2$ using a pretrained network (e.g., DINO), optionally upscaled for higher resolution; (iii) cross-view warping is performed by projecting the 3D points from one image into the viewpoint of the other, producing warped features $\hat{F}_1, \hat{F}_2$; (iv) cosine similarity between warped and original features is computed in both directions, yielding scores $S(I_1 \to I_2)$ and $S(I_2 \to I_1)$. The final MEt3R score is defined as:

$$\text{MEt3R}(I_1, I_2) = 1 - \tfrac{1}{2}\big(S(I_1 \to I_2) + S(I_2 \to I_1)\big),$$

where lower values indicate better multi-view consistency. Unlike pixel-based metrics, MEt3R is robust to view-dependent appearance changes (e.g., lighting), and can be applied pairwise across video frames to measure temporal 3D consistency. This makes it a suitable metric for evaluating panoramic video generation under curved camera paths.

**Camera Pose AUC@30.**  Camera Pose AUC@30 (Wang et al., 2025) measures the accuracy of predicted camera trajectories in generated videos. Relative Rotation Accuracy (RRA) and Relative Translation Accuracy (RTA) quantify angular errors in rotation and translation, respectively. For each threshold, the minimum of RRA and RTA is used, and the area under this accuracy-threshold curve (AUC) is computed. AUC@30 normalizes this area over thresholds $\theta \in [0°, 30°]$, reflecting the proportion of frames whose camera poses fall within a $30°$ tolerance. Higher values indicate more accurate and stable trajectory estimation.

For all tested videos, we resize each image to $1024 \times 576$ pixels and compare them with the ground truth videos at the same dimensions. For latent MSE of images, each image is resized to $299 \times 299$ pixels and processed through the Inception v4 model (Szegedy et al., 2017) to compute the score.

We present the mean and standard deviation across all test samples(except for FVD) in Table S4 and Table S5 as a summary of central tendency and variability of our method compared with the best baseline – GenEx for Table 1 and Table 2, respectively

|  | Polyline Paths | | | | Curved Paths | | | |
|---|---|---|---|---|---|---|---|---|
|  | LMSE↓ | LPIPS↓ | PSNR↑ | SSIM↑ | LMSE↓ | LPIPS↓ | PSNR↑ | SSIM↑ |
| GenEx | $0.046_{0.118}$ | $0.123_{0.049}$ | $24.190_{2.843}$ | $0.865_{0.035}$ | $0.113_{0.262}$ | $0.400_{0.093}$ | $17.110_{2.165}$ | $0.743_{0.046}$ |
| EvoWorld | $\mathbf{0.041}_{0.106}$ | $\mathbf{0.108}_{0.047}$ | $\mathbf{24.357}_{2.622}$ | $\mathbf{0.869}_{0.033}$ | $\mathbf{0.065}_{0.161}$ | $\mathbf{0.167}_{0.065}$ | $\mathbf{22.026}_{2.218}$ | $\mathbf{0.826}_{0.038}$ |

Table S4: Quantitative comparison for different camera trajectories in Unity. Metrics are reported as mean$_{std}$ for latent MSE, LPIPS, PSNR, and SSIM under Polylines and Curved Paths. Except for PSNR in the Curved Path setting, EvoWorld consistently shows lower standard deviation than Genex across all metrics.

|  | LMSE↓ | LPIPS↓ | PSNR↑ | SSIM↑ |
|---|---|---|---|---|
|  | mean$_{std}$ | mean$_{std}$ | mean$_{std}$ | mean$_{std}$ |
| *Unity* | | | | |
| GenEx | $0.184_{0.401}$ | $0.517_{0.081}$ | $14.558_{1.270}$ | $0.714_{0.037}$ |
| EvoWorld | $\mathbf{0.173}_{0.384}$ | $\mathbf{0.494}_{0.086}$ | $\mathbf{15.495}_{1.456}$ | $\mathbf{0.730}_{0.038}$ |
| *UE* | | | | |
| GenEx | $0.192_{0.476}$ | $0.594_{0.048}$ | $12.043_{1.598}$ | $0.420_{0.045}$ |
| EvoWorld | $\mathbf{0.148}_{0.357}$ | $\mathbf{0.416}_{0.060}$ | $\mathbf{16.630}_{1.903}$ | $\mathbf{0.500}_{0.048}$ |
| *Indoor* | | | | |
| GenEx | $\mathbf{0.218}_{0.480}$ | $0.665_{0.072}$ | $12.164_{2.416}$ | $0.545_{0.127}$ |
| EvoWorld | $\mathbf{0.218}_{0.492}$ | $\mathbf{0.624}_{0.080}$ | $\mathbf{12.990}_{2.608}$ | $\mathbf{0.559}_{0.122}$ |
| *Real* | | | | |
| GenEx | $0.191_{0.462}$ | $0.606_{0.090}$ | $12.743_{2.440}$ | $0.383_{0.105}$ |
| EvoWorld | $\mathbf{0.183}_{0.426}$ | $\mathbf{0.583}_{0.095}$ | $\mathbf{13.439}_{2.343}$ | $\mathbf{0.396}_{0.111}$ |

Table S5: Quantitative comparison with mean$_{std}$ for MSE, LPIPS, PSNR, and SSIM across different domains and models. For MSE, EvoWorld has a lower standard deviation in 3 out of 4 domains. For the other metrics (LPIPS, PSNR, and SSIM), EvoWorld consistently exhibits higher standard deviation than Genex.

**Analysis on statistics**. In the single-clip generation setting (Table S4), EvoWorld shows a lower standard deviation across four metrics compared to GenEx. In the recursive generation setting (Table S5), EvoWorld exhibits a bit higher variability in LPIPS, PSNR, and SSIM, while maintaining lower variability in MSE. These trends in standard deviation are consistent and within a normal range, suggesting the reliability of the results. MSE shows a larger standard deviation compared with other metrics, which is expected due to its quadratic sensitivity to outliers. It amplifies localized errors such as misalignment, leading to greater variance. This suggests a long-tail distribution in video prediction. Overall, EvoWorld achieves better performance than GenEx with stable and interpretable variance.

## A.9 ASSETS LICENSES

### A.9.1 GAME ENGINES

**Unreal Engine 5**: Free license for education. https://www.unrealengine.com/en-US/license

**Unity**: Free license for education. https://unity.com/products/unity-education-grant-license

### A.9.2 OPEN-SOURCED MODELS

**SVD**: Proper use under the "RESEARCH & NON-COMMERCIAL USE LICENSE"

StableVideoDiffusion1.1License

**WAN2.1**: Apache License

https://huggingface.co/Wan-AI/Wan2.1-I2V-14B-480P

**LTX-Video**: The research purpose is under the permitted license https://huggingface.co/Lightricks/LTX-Video/blob/main/ltx-video-2b-v0.9.license.txt **Nvidia Cosmos**: Apache License. https://github.com/NVIDIA/Cosmos/blob/main/LICENSE

**CogVideo-X**: Apache License. https://github.com/THUDM/CogVideo/blob/main/LICENSE

### A.9.3 OPEN-SOURCED DATASET

**Indoor HM3D dataset**: Free for research. https://aihabitat.org/datasets/hm3d/

**Indoor MP3D dataset**: Permitted for academic use. https://kaldir.vc.in.tum.de/matterport/MP_TOS.pdf

### A.9.4 OUR NEW ASSETS

Our newly collected dataset will use the CC BY 4.0 license.

### A.10 USE OF LANGUAGE MODELS

To improve clarity and readability, we used large language models (LLMs) as an assistant to refine grammar, polish phrasing, and consolidate sections of the manuscript. Besides, we use GPT-4o as a vision-language model to evaluate spatial consistency on the downstream tasks, which is described in detail in Section A.6. The technical content, main experiments, results, and conclusions remain our own, and all scientific claims were carefully validated by the authors. LLM assistance on language editing did not influence the substance of the work.

