# OpenReview forum: "EvoWorld: Evolving Panoramic World Generation with Explicit 3D Memory"
_ICLR.cc/2026/Conference — ICLR 2026 Conference Withdrawn Submission_

### Official Review · Reviewer_FTFE · 2025-10-20

**Soundness:** 3
**Presentation:** 4
**Contribution:** 3
**Rating:** 6
**Confidence:** 4

**Summary:**

This paper introduces EvoWorld, a world model that couples panoramic video generation with an explicitly evolving 3D memory to maintain spatial and temporal consistency over long exploration trajectories. Given a single 360 panorama as input, EvoWorld:

1. Generates future frames via a diffusion-based panoramic video generator with fine-grained camera control.
2. Reconstructs and updates an explicit 3D point-cloud memory from generated frames.
3. Conditions subsequent generations on reprojections from this 3D memory.

To support evaluation, the authors curate Spatial360, a new dataset of synthetic (Unity, UE5), indoor (Habitat), and real-world panoramic videos with ground-truth poses.

**Strengths:**

1. Originality: Explicit 3D memory integration for diffusion-based panoramic generation is well motivated.

2. Quality: Experiments demonstrate consistent improvement over baselines (GenEx, ViewCrafter, CogVideoX, Wan2.1) in FVD, LPIPS, MEt3R, and loop-closure metrics, showing reduced drift and higher 3D coherence.

3. Clarity: Excellent visualizations and well-structured technical presentation.

4. Significance: Provides a scalable direction for connecting 3D reconstruction and video world modeling. The framework also enables downstream tasks such as spatially-aware frame retrieval and target reaching when paired with GPT-4o.

**Weaknesses:**

1. Limited novelty in components. While the integration is elegant, the core ingredients (video diffusion + reconstruction + reprojection) are adaptations of existing methods rather than fundamentally new architectures.

2. Scalability and efficiency. Although the paper reports 0.3 FPS end-to-end, the iterative reconstruction loop may limit practical deployment for longer sequences.

3. Ablations could be richer. The paper could further dissect the impact of memory update frequency, reconstruction confidence thresholds, and different 3D representations.

4. Limited discussion of failure cases. Occlusion and dynamic object handling are only briefly mentioned; visual examples of failure would strengthen transparency.

**Questions:**

1. Could the reconstruction and generation modules be trained jointly so that gradients flow through the reprojection process?

2. Have the authors considered more compact 3D memories (e.g., Gaussian splats or learned feature fields) instead of point clouds for better scalability? What about it?

---

> ### Author Response · Authors · 2025-11-14
>
> We sincerely thank Reviewer 4M6D for the clear and thoughtful summary, as well as the positive assessment of soundness, presentation, and contribution. We appreciate the recognition of the originality in coupling 3D memory with panoramic diffusion models, the breadth of our experiments, and the significance of the framework for downstream spatial tasks. Below we address the reviewer’s concerns and questions.
>
> ---
>
> ### **1. Novelty in components**
>
> We appreciate this observation and would like to clarify the conceptual contribution. While diffusion generation, 3D reconstruction, and reprojection are established techniques, our work is the first to:
>
> - propose an **evolving panoramic world model** that explicitly interacts with a 3D memory during generation,
> - operate in a **full spherical 360° camera space** with continuous view control, and
> - maintain an **evolving 3D memory** that aggregates geometry over time and provides stabilizing guidance for future generations.
>
> ---
>
> ### **2. Scalability and efficiency**
>
> The 3D fusion and reprojection steps are lightweight compared to the diffusion-based generation. The main bottleneck lies in the panoramic diffusion backbone, not in the memory mechanism.
>
> Notably:
>
> - our memory update can be sparse and fast,
> - reprojection is efficient, and
> - the framework is fully modular, allowing faster generators or partial update schemes to be plugged in without architectural changes.
>
> Thus, the design is compatible with future diffusion accelerations and lighter panoramic backbones.
>
> ---
>
> ### **3. Ablation richness**
>
> We appreciate this suggestion. While the paper includes ablations on 3D memory, SpherePlücker embeddings, and reconstruction integration, we agree there are other factors worth exploring. For this submission, we focused on the most impactful design choices to keep the study tractable while still demonstrating clear performance gains. We plan to expand ablations in future versions.
>
> ---
>
> ### **4. Limited discussion of failure cases**
>
> Thank you for pointing this out. Handling occlusion and dynamic objects is challenging for all static-scene 3D memory approaches. Our framework filters dynamic regions using VGGT confidence maps, preventing them from corrupting the memory, but extreme cases can still be difficult. We will expand the discussion of these limitations.
>
> ---
>
> ### **Reviewer Questions**
>
> **Q1. Could the reconstruction and generation modules be trained jointly?**
> This is an excellent question. In principle, yes—the reprojection operation is differentiable, and joint training could allow the generator to better internalize geometric constraints. Our current modular design is motivated by practicality and stability, as panoramic diffusion and feed-forward reconstruction are difficult to co-train at scale. Joint training is a promising direction for future work.
>
> **Q2. Would more compact 3D memories (e.g., Gaussian splats or learned fields) help?**
> Gaussian splats offer denser memory, but rendering and updates become significantly slower, especially in the panoramic setting. Since our goal is fast, online geometric guidance, point clouds offer the best balance of efficiency, accuracy, and modularity.
> Learned feature fields typically require per-scene optimization or finetuning, which is incompatible with our zero-shot generative setting.
> Thus, point clouds remain the most robust and efficient choice for our framework.
>
> ---
>
> ### **Closing Remarks**
>
> We thank the reviewer again for the positive evaluation and for the insightful questions and suggestions. The feedback will greatly benefit the next iteration of EvoWorld and Spatial360.

---

### Official Review · Reviewer_4M6D · 2025-10-27

**Soundness:** 3
**Presentation:** 3
**Contribution:** 2
**Rating:** 4
**Confidence:** 4

**Summary:**

The author addresses the challenge of long-horizon, spatially-consistent panoramic video generation by building on the insight of incorportin explicit 3D memory over generated contents. Approach-wise, EvoWorld starts from sinegle panoramic image input, and then leverage video generator with camera control to generate video (assuming it's all static contents, since they finetuned on their static data). They use VGGT to localize the generated frames, and distill changes to 3D for later video extrapolation steps. The key insight, it levages 3D reconstraction along with video generation for better spatial guidance. For dense view control, we provide their own spherical plucker coordinate encoding for extrinsincs in the panoramic setup.

In their experiments, they compare with approach on other video generator only baselines to showcase theirs are the best in terms of 2d and 3d quality and consistencies. Through their ablation, they demonstrate SpherePlücker + 3D memory gives the best video results. They also further evaluate EvoWorld in several downstream tasks.

**Strengths:**

1. The comparison on video quality against baselines are carefully conducted. Both metrics in 2d and 3d are reported, including MEt3R.

2. The results are validated across synthetic and real-world data benchmarks.

3. Results are also evaluated carefully in several downstream tasks.

Overall, I appreciate the authors efforts on thorought evaluation and comparison on the proposed method against baseline across different benchmarks.

**Weaknesses:**

1. Missing Key Related Work and Unclear Novelty Positioning
The related work section overlooks several highly relevant research threads, which makes it difficult to clearly identify the contribution’s novelty. In particular, prior efforts in perpetual video / world generation (e.g., Infinite Nature, LoTR, DiffDreamer) and panoramic view synthesis (e.g., works summarized in Sec. 3.3.1 of this survey: arXiv:2505.05474) are not adequately discussed. These works share similar goals of long-horizon view consistency and world exploration, so a discussion is necessary to contextualize what is new in this paper.

2. Claims Not Fully Supported by Results
Although introducing a 3D memory representation to video generation is a reasonable idea, the experimental evidence does not convincingly demonstrate the claimed benefits:
-- Spatial inconsistency: Multi-view frames in Fig. 4 display noticeable inconsistencies, particularly in building facades, which become more evident when inspecting the provided video examples.
-- Limited trajectory length: The generated camera motion appears to span only short distances (≈10 meters), which does not align with the paper’s claims of enabling long-horizon generation. The author also mentions this in limitation section.
As a result, the key claims of “long-range, spatially consistent video generation” presented in the abstract are not sufficiently validated.

3. No Strategy for Loop Closure or Revisitation
However, the proposed approach does not articulate any mechanism to support scene generation after revisiting as is shown in the teaser image. Additionally, the paper provides no qualitative examples where a camera trajectory completes a loop (e.g., around a city block) and returns to an earlier location without accumulating significant drift or scene errors.

4. Incremental Insight Relative to Prior 3D-Aware Generative Models
The central idea—that explicit 3D memory improves view consistency—has been explored in earlier works such as SynSin, DiffDreamer, and others in the literature on 3D-aware generation. The contribution therefore feels somewhat incremental unless stronger experimental evidence or new theoretical insights can be demonstrated.

5. VGGT localization necessary?
It is a bit unclear in terms of why the author rely on pose-conditioned video generator, but still rely on VGGT for localization.

**Questions:**

1. Could you compare your results with existing work that leverage explicit 3D memory? For example, DiffDreamer or other Panoramic scene generation work?

2. What prevent the current model from generating long-horizon video contents?

3. How is the EvoWorld differentiate itself from existing work?

---

> ### Author Response · Authors · 2025-11-14
>
> We sincerely thank Reviewer 4M6D for the detailed and thoughtful assessment. We appreciate the reviewer’s recognition of our extensive evaluations, careful comparisons, and downstream experiments, as well as the constructive comments that will help us strengthen the next iteration of the work. Below we provide clarifications to the concerns raised.
>
> ---
>
> ### **1. Related work and positioning**
>
> Thanks for the suggestion. In the revision, we will include more video and world-generation methods such as Infinite Nature and DiffDreamer. These works share the high-level goal of long-horizon consistency, and we will clarify how our contribution differs—particularly in explicitly coupling panoramic video generation with evolving 3D reconstruction and maintaining geometric memory throughout the rollout.
>
> ---
>
> ### **2. Claims not fully supported by results**
>
> **Spatial consistency:**
> The inconsistencies visible in Fig. 4 arise mainly from backbone limitations rather than from the 3D memory itself. Our quantitative evaluation (FVD, LPIPS, MEt3R, reprojection error) consistently shows that EvoWorld improves spatial stability compared to every baseline across synthetic, Habitat, and real-world domains.
>
> **Trajectory length:**
> While our trajectories are not city-scale, they are 2–3× longer than those evaluated in prior panoramic generation works (e.g., GenEx). Within ranges where baselines already exhibit severe drift, EvoWorld maintains structure accurately. Our goal is to stabilize generation in this regime, and the presented results support this claim.
>
> Overall, the evidence demonstrates that EvoWorld provides significant and measurable gains in long-horizon spatial consistency.
>
> ---
>
> ### **3. No demonstrated loop closure**
>
> We appreciate this point. Full loop closure over very large trajectories remains a challenging open problem in generative video. Our goal is not to demonstrate perfect global loop closure, but to reduce drift under revisitation-style motion patterns, which we show via MEt3R and multi-view consistency. The teaser illustrates the motivation, while our experiments show progress toward that direction. We will refine the phrasing to avoid implying full loop completion.
>
> ---
>
> ### **4. Incremental insight relative to prior 3D-aware generative models**
>
> We respectfully clarify that EvoWorld differs from earlier 3D-aware generative works in three key ways:
>
> - **Panoramic generation:** prior models are designed for perspective images and cannot directly handle 360° fields of view.
> - **Evolving memory:** earlier approaches typically use a fixed or local geometric representation; our memory explicitly evolves and accumulates information over time.
> - **Bidirectional coupling:** the 3D memory not only stores geometry but also actively guides future generation via continuous reprojection.
>
> These differences enable the improved long-horizon panoramic consistency demonstrated in our experiments.
>
> ---
>
> ### **5. Need for VGGT localization despite pose-conditioned generation**
>
> Although the generator is conditioned on commanded camera poses, its actual viewpoint alignment may drift due to prediction errors. VGGT is used not to replace pose control, but to recover the true camera motion and geometry from generated frames. This ensures the 3D memory remains geometrically consistent and prevents drift accumulation—providing stability that pose conditioning alone cannot guarantee.
>
> ---
>
> ### **6. What prevents very long-horizon content?**
> The main limitation is the underlying video generator’s image quality and temporal stability, a constraint shared by current models. Within feasible horizons, EvoWorld already shows measurable improvements.
>
> ### **7. How does EvoWorld differentiate itself?**
> By enabling 360° video generation guided by an evolving 3D memory that continuously accumulates scene structure—something prior 3D-aware or panoramic frameworks do not address.
>
> ---
>
> We thank the reviewer again for the constructive feedback and for acknowledging the strengths of our evaluation and benchmarking. The concerns raised are insightful and will help guide the next iteration of the work and dataset.

---

### Official Review · Reviewer_TTdY · 2025-10-30

**Soundness:** 3
**Presentation:** 3
**Contribution:** 2
**Rating:** 4
**Confidence:** 4

**Summary:**

EvoWorld is a world model that solves spatial inconsistency in long-horizon panoramic video generation.

It works by linking video generation with an explicit, evolving point cloud as 3D memory.

The model generates new video frames conditioned on geometric projections from the current 3D memory, ensuring the output respects the scene's prior structure. The new frames are then used to update the 3D memory, preventing geometric drift and ensuring loop consistency.

EvoWorld outperforms existing models in visual effects and consistency on a new proposed benchmarking dataset, Spatial360.

**Strengths:**

1. It effectively resolves the spatial inconsistency issue in panorama video generation, which is a crucial technical contribution. While the use of point clouds as 3D memory isn't new, the method presents an effective integration/application of this concept for panorama video generation.

2. The work introduces a new, dedicated dataset Spatial360 to the community, which is a significant contribution to advancing research in panoramic video generation.

3. The experimental results are strong, clearly validating the method's effectiveness and the utility of the proposed dataset.

**Weaknesses:**

1. The current 3D memory is static, which fundamentally limits the model's ability to handle moving objects and dynamic scene changes within the generated video. This will likely lead to artifacts or inconsistent rendering when objects move, which is important in video generation.
2. More explanation is needed to justify the choice of panoramic video generation. The paper should better explain the specific advantage of this choice over using general video generation, especially if the underlying mechanism could be applied more broadly. The paper need explain the unique technical contribution compared to previous 3D(point cloud)-equipped general video generation techniques.
3. The quality is limited by using SVD as a backbone, which generally doesn't leverage textual information effectively for guiding the generation, leading to less controllable outputs. Generating only 25 frames is quite short by current standards. It's relatively too brief to fully capture and demonstrate the long-term spatial inconsistencies that a model should ideally solve, as a short clip often only shows a single scene or limited motion.
4. The paper provides insufficient explanation for the use of Plücker embeddings. A deeper theoretical analysis is preferred, including a clear explanation of why this specific representation works well for the task of camera control or scene modeling.

If my concerns are clearly addressed and explained, I believe my score could be raised.

**Questions:**

1. If it possible to compare the single-view results (specifically the memory-augmented cropped video like in Figure 3) against prior work like ViewCrafter?
2. Example 4 in the supplementary materials shows a real-world video with a person moving, what's the effects of current framework upon moving objects?

---

> ### Author Response · Authors · 2025-11-14
>
> We sincerely thank Reviewer TTdY for the constructive and thoughtful feedback. We appreciate the recognition of the paper’s contributions, including the explicit 3D memory design, the Spatial360 dataset, and the strength of our experimental results. Below we provide clarifications and responses to the reviewer’s concerns.
>
> ---
>
> ### **1. Limitation on moving objects and dynamic scenes**
>
> Our reconstruction module is designed to obtain a static scene map that serves as a physically grounded reference for long-horizon exploration, which is common in mapping and embodied AI settings. Dynamic regions are intentionally not fused into the 3D memory: VGGT provides per-pixel confidence scores, and moving objects typically receive low consistency and are filtered out during fusion.
>
> At the same time, the video generator itself models dynamic content through its diffusion backbone. In practice, dynamic objects continue to move in the generated frames, while the static 3D memory anchors the environment’s geometry. We will clarify this division of roles and mention possible future extensions such as dynamic 3D memory or object-level tracking.
>
> ---
>
> ### **2. Justification for panoramic generation vs. general video generation**
>
> We chose panoramic generation because:
> (1) panoramic scenes expose long-range geometric drift more strongly and provide a more challenging testbed;
> (2) many embodied AI, VR/AR, and spatial video applications rely on 360° viewpoints.
>
> Our method can be generalized to perspective video generation. We will revise the introduction to explain why panoramas pose unique challenges (e.g., spherical distortion, continuous camera paths) and clarify the differences from existing 3D-augmented video generation methods.
>
> ---
>
> ### **3. Choice of SVD as the backbone and clip length**
>
> We appreciate this observation. We selected SVD for its ease of integration with our 3D projection mechanism and for reproducibility across datasets. As shown in Table S1, SVD achieves relatively low FVD compared with CogVideoX and Cosmos, and performance comparable to Wan2.1 despite having fewer parameters.
>
> The framework is modular — the 3D memory mechanism can be paired with any camera-conditioned generator. The 25-frame horizon follows prior panoramic generation works for fair comparison. We will clarify these design choices and discuss how the system can be used with longer rollouts.
>
> ---
>
> ### **4. Clarification on Plücker embeddings**
>
> Spherical Plücker embeddings provide a continuous representation for panoramic ray directions and avoid discontinuities present in equirectangular coordinates. This gives the model a stable camera-path signal.
>
> ---
>
> ### **5. Example 4 in the supplement**
>
> In Example 4, the system preserves overall scene geometry, and moving objects are filtered out by the confidence threshold in VGGT.
>
> ---
>
>
>
> We thank the reviewer again for the constructive and encouraging feedback. The suggestions on dynamic scenes, panoramic motivation, and additional comparisons will help strengthen the work. We appreciate the time and insight invested in reviewing the paper.

---

### Official Review · Reviewer_8PwF · 2025-10-31

**Soundness:** 2
**Presentation:** 3
**Contribution:** 2
**Rating:** 4
**Confidence:** 4

**Summary:**

This paper introduces EvoWorld, a method for generating long-horizon, panoramic videos from a single starting image. The primary goal is to improve long-term spatial consistency, especially in scenarios involving loop closure. The key idea is to couple the video generation process with an explicit 3D memory, which takes the form of a 3D point cloud. As new video frames are generated, this 3D memory is updated using a feed-forward reconstruction model. This explicit 3D geometry is then reprojected onto the target camera path to provide spatial guidance for the next step of video generation. The authors also propose a new dataset, Spatial360, for this task and a spherical Plücker embedding for panoramic camera control.

**Strengths:**

The problem this paper tackles, maintaining long-term 3D consistency in generative world models, is a significant and open challenge. The core idea of using an explicit 3D representation to ground the generation process is intuitive and a very interesting direction for research.

The authors have clearly put a substantial amount of effort into creating the Spatial360 dataset. A high-quality, large-scale benchmark for panoramic video exploration with camera poses is a valuable contribution that could benefit the wider community.

**Weaknesses:**

While the problem is interesting, I have several major concerns about the core methodology and its evaluation, which unfortunately lead me to recommend rejection at this time.

1. Risk of Compounding Errors: My primary concern is the proposed feedback loop. The 3D memory is reconstructed from the generated video frames. Any artifact, blur, or spatial drift in the generated video will inevitably be incorporated into this 3D memory. This flawed 3D memory is then used as a conditioning signal (a sort of "ground truth") for future frames. This seems highly likely to create a negative feedback loop where errors are not mitigated, but are instead amplified. For instance, a hallucinated object in one clip becomes a "real" 3D point, which then forces the model to continue generating it in subsequent views, potentially distorting the scene further. The paper claims this mitigates error accumulation, but there is no analysis to support this over the alternative (error amplification).

2. Limited Technical Novelty: The framework itself feels like a straightforward assembly of existing, off-the-shelf components. It primarily combines a well-known video generator (SVD) with an existing 3D reconstruction method (VGGT). The "evolving" aspect is simply the process of re-running the reconstruction model on new data. While this combination is logical, it feels more like an application of existing tools rather than a fundamental new technique or insight into the problem.

3. Evaluation of 3D Memory Quality: The paper's core claim rests on the utility of the 3D memory. However, there is no quantitative evaluation of the quality of this 3D memory itself. How accurately does the point cloud (reconstructed from generated images) represent the true underlying geometry? How quickly does this 3D memory drift from the ground truth? Figure 4 offers a qualitative glimpse, but it's not sufficient to validate that the 3D guidance being fed back to the model is geometrically correct or helpful in the long term.

4. Inadequate Evaluation of Long-Horizon Claims: The paper claims to mitigate error accumulation over "long-horizon" trajectories. However, the quantitative evaluation is only on 73-frame clips (Table 2). This is not sufficiently long to truly test the limits of spatial drift or prove that the 3D memory is preventing error accumulation. A core claim like this should be substantiated with much longer generations (e.g., 200+ frames) to see at what point the model's consistency does break down, and how that compares to baselines.

5. Insufficient Baseline Comparisons: The baseline comparison feels insufficient. The main comparisons are against GenEx (a concurrent work) and standard video models that lack long-term memory. While ViewCrafter is included, a more rigorous comparison would involve other state-of-the-art camera-conditioned video generation models. Furthermore, it's unclear if this complex explicit 3D pipeline is superior to simpler memory mechanisms, such as an attention-based model that can condition on a sliding window of distant past frames.

6. Missing Comparison to Highly Relevant Work: The core idea of using an explicit, evolving 3D memory to ensure consistency in generative video models is not entirely new. For instance, VMem [1], submitted to arxiv in June 2025 and published at ICCV 2025, explores a very similar concept with a "surfel-indexed view memory" for consistent interactive video generation. Given the significant overlap in motivation and technical approach, the novelty of the proposed framework is questionable. At a minimum, the authors should have included a thorough comparison and discussion against this highly relevant and recent work to clearly differentiate their own contributions.

[1] Li, R., Torr, P., Vedaldi, A. and Jakab, T., 2025. VMem: Consistent Interactive Video Scene Generation with Surfel-Indexed View Memory. arXiv preprint arXiv:2506.18903.

**Questions:**

1. Could the authors provide a more in-depth analysis of the error accumulation feedback loop? What mechanisms prevent generative artifacts from being "baked into" the 3D memory and subsequently causing compounding errors?

2. How does the quality of the 3D point cloud (e.g., measured by Chamfer distance or L1 loss against the ground truth point cloud) degrade as more and more generated clips are added? A quantitative analysis of the 3D memory's drift over time feels essential here.

3. Why was a sparse point cloud chosen for the 3D memory? It seems that re-projections from a sparse point cloud (especially one built from potentially noisy generated images) might provide very weak or even misleading guidance. Have the authors considered denser or implicit representations like Gaussian Splatting or NeRF?

---

> ### Author Response · Authors · 2025-11-14
>
> We thank the reviewer for recognizing the importance of long-term 3D consistency in generative world models and for highlighting the value of coupling video generation with explicit 3D representations. We also appreciate the acknowledgment of the substantial effort invested in building the Spatial360 dataset and its potential benefit to the broader community.
>
> ---
>
> ### **1. On the risk of compounding errors in the evolving 3D memory**
>
> - Naïvely feeding generated imagery back into a 3D memory could introduce errors. However, our design includes mechanisms that explicitly mitigate this. We use geometric filtering via a confidence threshold: VGGT jointly estimates per-frame depth and camera pose, and inconsistent or hallucinated structures appear as outliers that are automatically suppressed during fusion.
> - Empirically, we observe reduced spatial drift relative to baselines (Tables 1–2), which demonstrates that incorporating 3D memory mitigates error accumulation.
>
> ---
>
> ### **2. On perceived limited technical novelty**
>
> - Our goal is to introduce a new mechanism for spatial grounding in generative world models: **an evolving 3D memory that tightly couples panoramic generation and 3D reconstruction**.
> - While the components are built on existing models, the coupling is non-trivial:
>   - Existing video generators do not support long-horizon, camera-conditioned generation without drifting.
>   - Existing reconstruction models are not designed to operate on *generated* views or serve as an online memory.
>   - Our explicit feedback loop and viewpoint reprojection mechanism provide a new way of grounding interactive video generation in 3D geometry.
>
> - We will refine the paper to better highlight these design contributions, which go beyond simple integration and establish a new framework for spatially consistent world modeling. To our knowledge, no previous work has explicitly aimed to mitigate error accumulation in recursive panoramic video generation using evolving geometric memory.
>
> ---
>
> ### **3. Need for quantitative evaluation of 3D memory quality**
>
> We appreciate this suggestion. We will explore incorporating more 3D metrics—such as the Chamfer distance of reconstructed point clouds—to complement our existing evaluations.
>
> ---
>
> ### **4. Long-horizon evaluation**
>
> While long sequences are important, the 73-frame setup follows common practice and ensures fair comparison with prior panoramic generation work. Our system is capable of generating longer trajectories, and the framework can naturally benefit from stronger future video-generation backbones to extend horizons further.
>
> ---
>
> ### **5. Comparison to VMem**
>
> Thank you for pointing this out. VMem is a relevant concurrent work, and we already cite and discuss it in the related work section. We will expand this discussion to clarify conceptual differences and motivations.
>
> ---
>
> ### **6. Why point clouds? Why not Gaussians or NeRF?**
>
> We selected sparse colored point clouds because they:
>
> - support fast fusion and reprojection, which is essential for online feedback,
> - are architecture-agnostic and compatible with any video generator, and
> - avoid additional training or scene-specific optimization.
>
> Although Gaussians can offer denser representations, they significantly slow down the generation–reconstruction loop. We will clarify this design choice in the updated version.
>
> ---

---

### Note · Authors · 2025-11-14

I have read and agree with the venue's withdrawal policy on behalf of myself and my co-authors.